# Improving Topic Modeling by Distilling Soft Labels
# from Language Models

**Raymond Li** [1]   **Amirhossein Abaskohi** [1]   **Chuyuan Li** [1]   **Gabriel Murray** [1]   **Giuseppe Carenini** [1]

## Abstract

Traditional neural topic models are typically optimized by reconstructing the document's Bag-of-Words (BoW) representations, overlooking contextual information and struggling with data sparsity. In this work, we introduce a novel topic model training framework by **D**istilling **S**oft **L**abels (DSL) from Language Models (LMs). To construct the contextually enriched reconstruction signals, we project the next token probabilities, conditioned on a specialized prompt, onto a predefined vocabulary, and train the topic models to reconstruct the soft labels using the LM hidden states. This produces higher-quality topics that are more closely aligned with the underlying thematic structure of the corpus. Extensive experiments demonstrate that DSL achieves substantial improvements in topic coherence and assignment accuracy over existing baselines. Additionally, we also introduce a retrieval-based metric, which shows that our approach significantly outperforms existing methods in identifying semantically similar documents, highlighting its effectiveness for retrieval-oriented applications.

## 1. Introduction

Topic modeling is used for uncovering the latent thematic structures in large text corpora. Under the probabilistic topic modeling paradigm (Blei et al., 2003), documents are modeled as mixtures of topics, and each topic can be represented by the most probable words to provide a human-interpretable summary. The document-level topic compositions are useful for a wide array of applications including information retrieval (Bai et al., 2018; Li et al., 2021), summarization (Nguyen et al., 2021; Zhang et al., 2022a), and exploratory data analysis (Zeng et al., 2019; Li et al., 2020).

Obviously some reporter for the Ottawa Sun got taken by an April Fools joke...probably started by someone with the Nordiques or the Bruins.

Like for example...who is **going** to reimburse the Flyers for the $15 **million** they **paid** to the Nordiques...like the Senators are **going** to get Lindros and $15 **million**. The Flyers sent the equivalent of 6 or 7 players (when you **include** the **draft** choices) to Quebec, and they are **going** to get only four back.

Some reporter was had **real** badly and someone must be **having** a **real good** laugh seeing as how the so much of the sports media has chosen to publicize this utter nonsense. ... ...

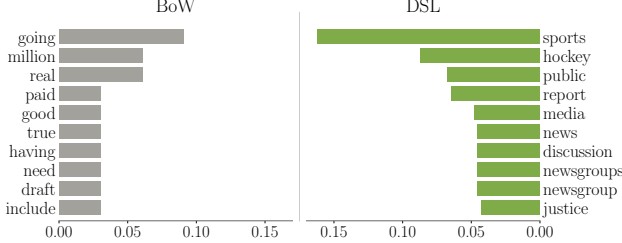

*Figure 1.* Comparison of BoW target (left) versus DSL target (right) for a document in the 20Newsgroups dataset. BoW can only assign mass to words that appear in the document (**bolded**), while the DSL target captures the underlying themes of the sports commentary even when the words don't explicitly appear in the document (e.g., "*sports*", "*hockey*").

Most existing neural topic models (Miao et al., 2017; Srivastava & Sutton, 2017; Bianchi et al., 2021a; Dieng et al., 2020) infer latent topics by reconstructing the document's **Bag-of-Words (BoW)** representation, which captures statistical word co-occurrences, but disregards compositional meaning and the rich contextual information crucial for deep semantic understanding of the underlying corpus. This limitation is especially pronounced for short-texts, where the sparse co-occurrence signals hinder the model's ability to learn coherent, high-quality topics (Qiang et al., 2022).

On the other hand, despite advances in Large Language Models (LLMs) (Brown et al., 2020; Touvron et al., 2023a;b; Grattafiori et al., 2024), they still struggle to effectively model large corpora due to context-length limitations and long-range dependency challenges. Consequently, prior work has primarily used LLMs to evaluate topic quality (Stammbach et al., 2023; Rahimi et al., 2024) or augment documents to alleviate data sparsity (Akash & Chang, 2024). Studies that use LLMs for topic modeling typically require

[1]University of British Columbia, Vancouver, Canada. Correspondence to: Raymond Li <raymondl@cs.ubc.ca>.

*Proceedings of the 43rd International Conference on Machine Learning*, Seoul, South Korea. PMLR 306, 2026. Copyright 2026 by the author(s).

**complex and costly pipelines** that are impractical for larger corpora (Mu et al., 2024; Pham et al., 2024; Yang et al., 2025b). Moreover, prompting-based approaches **do not provide probabilistic topic distributions**, a key advantage of neural topic models such as Latent Dirichlet Allocation (LDA) (Blei et al., 2003).

Following recent progress in Small Language Models (SLMs), which combine strong semantic performance with substantially lower computational cost and latency (Wang et al., 2025; Belcak et al., 2025), we propose a simple and efficient paradigm for topic modeling by **distilling soft label** (DSL) distributions from the LSM into probabilistic topic models. Specifically, using a simple prompting strategy (e.g., "*Generate a single word label that captures the underlying theme of the document.*"), SLM is instructed to generate a soft label target encoding the overall document theme. We mapping the probabilities of the next completion token immediately following the prompt to a soft label distribution over the predefined topic modeling vocabulary. By training the neural topic model to reconstruct these dense, semantically-rich soft targets, it learns to infer topics that are more contextually coherent and semantically meaningful. Furthermore, we use the last hidden state representations of the language models as the contextualized inputs for the documents, which provide the necessary context for reconstructing the soft targets. From the perspective of implicit Bayesian inference in language models (Xie et al., 2022; Wang et al., 2023; Li et al., 2025), DSL can be viewed as projecting the LM's implicit posterior predictive distribution over theme-relevant words onto the structured hypothesis space of topic models, yielding explicit and interpretable document-topic and topic-word distributions.

In summary, our contributions are[1]: (1) We propose a novel training paradigm for neural topic modeling that uses semantically grounded soft label distributions derived from a language model as the reconstruction objective; (2) We demonstrate the effectiveness of our approach through extensive experiments on three datasets, achieving substantial improvements over baseline methods in both topic quality and assignment accuracy; (3) We introduce a retrieval-based evaluation metric for topic modeling, showing that our learned document-topic representations substantially improve topic-guided document retrieval across datasets.

## 2. Related Work

### 2.1. Neural Topic Models

Built upon the generative probabilistic framework described by Latent Dirichlet Allocation (LDA) (Blei et al., 2003), the LDA family of neural topic models integrate deep neural network components through the variational autoencoder (VAE) framework (Kingma & Welling, 2014), using an encoder to approximate the posterior with a variational distribution $q(z|x)$ and a decoder to reconstruct the document's BoW representation from sampled topic proportions $\theta$. For example, Miao et al. (2017) mapped documents into continuous latent representation by using the encoder to predict the mean and variance of a Gaussian posterior, while ProdLDA (Srivastava & Sutton, 2017) replaced the mixture-of-multinomials decoder of LDA with a product of experts architecture where document-topic vectors are sampled from a logistic-normal prior. Another line of work has leveraged the semantic information of pre-trained embeddings (Fang et al., 2024; Abaskohi et al., 2025), examples include the Embedding Topic Model (ETM) (Dieng et al., 2020), which represents topics and words in a shared embedding space, where the document is reconstructed via the inner product between the topic and the initialized word embedding. Building on ProdLDA, CombinedTM (Bianchi et al., 2021a) concatenated the BoW document input with contextualized Sentence-BERT (SBERT) embeddings (Reimers & Gurevych, 2019), while ZeroShotTM (Bianchi et al., 2021b) replaced the input entirely with SBERT embeddings.

Alternatively, clustering-based methods (Zhang et al., 2022b) congregate contextualized embeddings to semantically similar word or document clusters. For example, Sia et al. (2020) performed clustering on word embeddings and obtain top words using weighing and re-ranking, while BERTopic (Grootendorst, 2022) performed clustering on document embeddings and use a class-based TF-IDF method to label each topical cluster. While these methods often achieve high coherence, clustering algorithms often have non-linear complexity and cannot efficiently scale-up to large corpora.

Finally, several approaches have leveraged optimal transport (OT) to regularize or align topic models in the embedding space (Nan et al., 2019; Zhao et al., 2021; Wang et al., 2022). Recent works include ECRTM (Wu et al., 2023), which augments VAE-based neural topic models with an embedding clustering regularization term to prevent topic collapse by separating topic embeddings, and FASTopic (Wu et al., 2024a), which employs an OT-based transport plan to align documents, topics, and words in a shared embedding space while reconstructing the BoW representation of documents.

### 2.2. LLMs for Topic Modeling

Recent studies have incorporated LLMs in topic modeling research. One line of work used LLMs for the evaluation of topic models, such as rating the quality of the topics (Stammbach et al., 2023; Rahimi et al., 2024) and their document representations (Yang et al., 2025a). Another line of work focused on improving short-text topic modeling by

---

[1]The implementation of our work is available at https://github.com/raymondzmc/dsl-topic-models

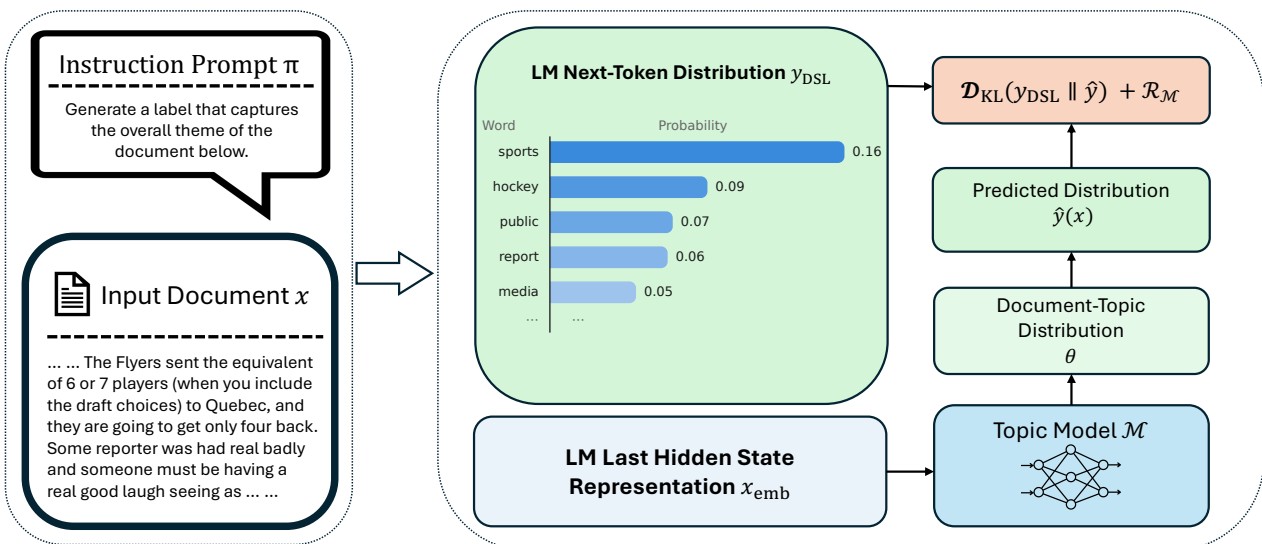

*Figure 2.* Overview of the proposed DSL framework. Given an input document $x$ and a thematic instruction prompt $\pi$, the language model produces a prompt-conditioned next-token distribution, which is projected onto the topic-model vocabulary to form the semantic soft target $y_{\text{DSL}}$. The LM final hidden-state representation $x_{\text{emb}}$ is used as the document representation for the topic model $\mathcal{M}$, which infers document-topic proportions $\theta$ and predicts a vocabulary-level distribution $\hat{y}(x)$. Training distills the LM-induced target into the structured topic model by minimizing $D_{\text{KL}}(y_{\text{DSL}} \| \hat{y})$ together with the model-specific regularization term $\mathcal{R}_{\mathcal{M}}$.

using LLMs to expand the original text to obtain a denser BoW representation (Akash & Chang, 2024). More recently, studies have leveraged the capabilities of LLMs to generate topics directly. For example, Mu et al. (2024) proposed prompt-based topic extraction framework where topics are directly generated, refined, and summarized through iterative prompting. Similarly, TopicGPT (Pham et al., 2024) focused on topic assignment using a pipeline that prompts the LLM to generate topics for each document, before refining the topics and assigning the most probable topic to each document. While the experimental results for these works demonstrated improved interpretability and topic assignment accuracy, these methods do not form a fully probabilistic framework since topics are represented by the natural language description and topic assignments are binary decisions. This limits the ability to model topic uncertainty and capture topic distribution necessary for downstream applications. Finally, Yang et al. (2025b) first train a neural topic model with BoW targets before using a LLM to refine word distributions of already learned topics using an optimal-transport alignment loss. This approach is complementary to our proposal as we focus on obtaining a higher quality topic model using the enhanced reconstruction target.

## 3. Method

Figure 2 provides an overview of our method. We describe the main components in the following subsections: reconstruction targets (§3.1), the loss function and training objective (subsection 3.2), the topic model input (§3.3), and our

Bayesian interpretation of the framework (§3.4).

### 3.1. Reconstruction Targets

Following the standard practice for topic modeling (Hoyle et al., 2021), we first pre-process the corpus to build a fixed vocabulary set $V$ from the top-$|V|$ most frequent words after tokenization and stop-word removal. Let $\pi$ denote the thematic instruction prompt. For each document $x$, we construct a prompt-conditioned input by pairing $x$ with $\pi$, which instructs the LM to predict a concise label for the document's underlying theme (e.g., "Generate a single-word label capturing the document theme."). Rather than decoding the label, we take the next-token logits $\ell_{\text{LM}}(x, \pi)$ immediately following the prompt, where we retain only the entries corresponding to vocab subset $V$, yielding $\ell_V(x, \pi) \in \mathbb{R}^{|V|}$. Finally, we compute the temperature-scaled softmax distribution as the semantically-grounded **DSL** targets (Equation 1).

$$y_{\text{DSL}}(x, \pi) = \text{softmax}\left(\frac{\ell_V(x, \pi)}{\tau}\right) \qquad (1)$$

Unlike BoW targets, which assign nonzero mass only to words that explicitly appear in the document, the DSL target $y_{\text{DSL}}(x, \pi)$ is a dense distribution over $V$ that can assign probability mass to words absent from the document but thematically relevant to it (Figure 1).

## 3.2. Distilling Soft Label Targets

DSL is a training paradigm for topic models that learn through vocabulary-level reconstruction. We first write this broad class of vocabulary-reconstruction neural topic models in a unified form. These models learn topics by predicting a document-conditioned predictive distribution $\hat{y}_\psi(x)$ over the vocabulary $V$ and training it against a re-construction target. This view includes VAE-based neural topic models such as ProdLDA, as well as recent models that retain a BoW reconstruction component while adding embedding-based regularization, such as ECRTM (Wu et al., 2023) and FASTopic (Wu et al., 2024a).

For a topic model $\mathcal{M}$ with trainable parameters $\psi$, the standard objective can be written abstractly as minimizing a reconstruction term together with model-specific regularization (Equation 2).

$$\mathcal{L}(x) = \mathcal{D}_{\text{recon}}\big(y_{\text{BoW}}(x), \hat{y}_\psi(x)\big) + \mathcal{R}_\mathcal{M}(x; \psi) \quad (2)$$

Here, $y_{\text{BoW}}(x)$ is the normalized BoW representation of document $x$, reconstruction term $\mathcal{D}_{\text{recon}}(\cdot)$ is typically the negative log-likelihood (NLL) between normalized BoW representation $y_{\text{BoW}}(x)$ and predicted distribution $\hat{y}_\psi(x)$, and $\mathcal{R}_\mathcal{M}$ collects all model-specific terms that are not the reconstruction objective. For instance, $\mathcal{R}_\mathcal{M}$ corresponds to the latent-prior regularizer in ProdLDA, the embedding-clustering regularizer in ECRTM, and the transport-based regularization terms in FASTopic.

$$\mathcal{L}_{\text{DSL}}(x) = \lambda D_{\text{KL}}\left(y_{\text{DSL}}(x, \pi) \parallel \hat{y}_\psi(x)\right) + \mathcal{R}_\mathcal{M}(x; \psi) \quad (3)$$

The DSL objective is presented in Equation 3. At the objective level, DSL replaces the sparse BoW reconstruction target with the LM-induced semantic target $y_{\text{DSL}}(x, \pi)$ from Equation 1, while preserving the model-specific regularization terms. The topic model learns to reconstruct the language model distribution via KL divergence similar to the knowledge distillation loss (Hinton et al., 2015). The optional hyperparameter $\lambda$ can be used to balance DSL reconstruction and regularization. A formal justification of the KL-based reconstruction objective is provided in Appendix J.

## 3.3. Hidden State Inputs

Motivated by prior work that uses pre-trained contextualized embeddings to improve document representations (Bianchi et al., 2021a; Fang et al., 2024), we use the LM hidden state as the input representation for the topic model. Let $x_{\text{emb}} = h_{\text{LM}}(x, \pi)$ denote the final-layer hidden state at the last position of the prompt-conditioned input. For autoregressive LMs, this hidden state is passed to the LM head to produce

the next-token logits, making $x_{\text{emb}}$ naturally aligned with the DSL target over $V$.

## 3.4. Bayesian Interpretation

Learning an explicit topic model from the LM-induced target from Equation 1 provides a natural Bayesian interpretation. Prior work has interpreted language models as implicit Bayesian predictors over latent concepts during next-token prediction (Xie et al., 2022; Wang et al., 2023). Under this framework, when conditioned on both the instruction prompt and the input document $x$, LM can be viewed as implicitly inferring a posterior distribution over latent concepts $c$, where $c$ acts as an approximate sufficient statistic for the posterior information contained in the prompt-conditioned document context (Wang et al., 2023). The projected next-token distribution $y_{\text{DSL}}(x, \pi)$ over $V$ can thus be interpreted as an implicit posterior predictive distribution over theme-relevant vocabulary words. For each vocabulary word $v \in V$, this posterior predictive interpretation can be written as Equation 4, where the dependence on the instruction prompt is omitted for notational simplicity.

$$y_{\text{DSL}}(v \mid x, \pi) \approx \int p_{\text{LM}}(v \mid c)\, p_{\text{LM}}(c \mid x, \pi)\, dc \quad (4)$$

Under our proposed DSL framework, the topic model infers an explicit probabilistic topic representation from this implicit and unstructured posterior predictive signal.

$$\mathcal{P}_\mathcal{M}(x_{\text{emb}}) = \{\hat{y}_\psi(\cdot \mid x_{\text{emb}}) : \psi \in \Psi_\mathcal{M}\} \subseteq \Delta^{|V|-1} \quad (5)$$

For a topic model $\mathcal{M}$ with trainable parameters $\psi$, Equation 5 denotes the family of vocabulary-level distributions that $\mathcal{M}$ can produce for the document representation $x_{\text{emb}}$. Unlike the LM-induced posterior predictive signal, each element of $\mathcal{P}_\mathcal{M}$ is constrained by the topic-model structure. For example, for VAE-based topic models, $\hat{y}_\psi(x)$ is induced through a low-dimensional topic bottleneck that represents each document using document-topic proportions. DSL can therefore be interpreted as distilling the LM's implicit Bayesian prediction over latent concepts, observed through the posterior predictive target $y_{\text{DSL}}(x, \pi)$, into an explicit topic model with interpretable document-topic and topic-word distributions. Concretely, the KL reconstruction term in Equation 3 implements this conversion by matching the topic model prediction $\hat{y}_\psi(x)$ to the LM-induced target $y_{\text{DSL}}(x, \pi)$ within the structured family $\mathcal{P}_\mathcal{M}(x_{\text{emb}})$.

## 4. Experiments

### 4.1. Settings

**Metrics** To start, we evaluate the intrinsic quality of the topics through their *coherence* and *diversity*, which have

shown to be directly correlated with the interpretability of the topics (Dieng et al., 2020; Li et al., 2023). For topic *coherence*, we use the widely adopted Coherence Value ($C_V$) (Röder et al., 2015) and the recently proposed **LLM** ratings (Stammbach et al., 2023). For topic *diversity*, we use the Inversed Rank-Biased Overlap (**I-RBO**) score (Terragni et al., 2021b; Bianchi et al., 2021a). Additionally, following prior studies (Poursabzi-Sangdeh et al., 2016; Hoyle et al., 2022; Pham et al., 2024), we evaluate topical *assignment accuracy* by measuring how well the most probable topic assignments align with the ground-truth labels through the **Purity** score (Zhao & Karypis, 2001). Lastly, we propose a retrieval-based metric to assess the effectiveness of the full topical distribution for finding semantically related documents by retrieving top-$N$ documents based on the KL-divergence of topic distribution, and report **Precision** as the proportion of retrieved documents that share the same ground-truth label as the query.

**Datasets** We train and evaluate our models on three publicly available text classification datasets with ground-truth labels: 20NewsGroup, StackOverflow (Xu et al., 2015), and TweetTopic (Antypas et al., 2022). For TweetTopic, we filter examples with single labels in order to compute the **Purity** score. Note that both StackOverflow and Tweet-Topic are short-text datasets that traditional topic models often struggle with. For all datasets, we restrict $V$ to vocabulary words that correspond to single tokens under the LM tokenizer ($98\%$ of words) for efficiency, and set vocab size $|V| = 2000$ following settings from prior work (Bianchi et al., 2021a). In Appendix G, we show the performance gains from DSL is consistent for different values of $|V|$. In practice, this parameter can be adjusted or adapted to specific domains without modifying the proposed method.

### 4.2. Baselines

We compare our approach against a variety of topic modeling baselines to demonstrate the benefits of our proposed method[2]. Specifically, we choose the following five models from the LDA-family: **LDA** (Blei et al., 2003), **ProdLDA** (Srivastava & Sutton, 2017), **CombinedTM** (Bianchi et al., 2021a), **ZeroShotTM** (Bianchi et al., 2021b), **ETM** (Dieng et al., 2020).

Additionally, we also compare with the clustering-based model **BERTopic** (Grootendorst, 2022), as well as the more recent models **ECRTM** (Wu et al., 2023) and **FASTopic** (Wu et al., 2024a). In our DSL experiments, our method adopts the ProdLDA, ECRTM, and FASTopic architecture without modifications.

---

[2]Implementation and hyperparameter details of baselines are provided in Appendix C.

### 4.3. Hyperparameters

Our method uses the identical hyperparameter settings as the baseline models ProdLDA, ECRTM, and FASTopic[3]. For model-specific parameters, we use a temperature of $\tau = 3$, and a loss weight $\lambda = 1e^3$ to balance the reconstruction term with the regularization term. For efficiency, we use five instruction-tuned Small Language Models (SLMs) to compute the soft label targets for training the topic models. Specifically, we use `Llama3.1-8B-Instruct` and `Llama3.2-1B-Instruct` from the Llama family (Grattafiori et al., 2024), `Phi-3-mini` (Abdin et al., 2024), as well as the more recent `Qwen-3.5-0.8B` (Qwen Team, 2026) and `ERNIE-4.5-0.3B` (Baidu-ERNIE-Team, 2025) both of which have been shown to be highly capable in instruction following. For baselines requiring contextualized embeddings (i.e., ZeroShotTM, CombinedTM, BERTopic, FASTopic), we use `GTE-large-en-v1.5` (Zhang et al., 2024): a strong-performing dense encoder model (0.4B) which has a similar size to our smallest SLM (0.3B).

### 4.4. Automatic Topic Evaluation Results

In Table 1, we present the main experimental results on the 3 datasets using automatic topic evaluation metrics by averaging over 4 numbers of topics ($K = 25, 50, 75, 100$)[4]. We apply DSL to the three topic models (**ProdLDA**, **ECRTM**, and **FASTopic**) and five SLMs spanning four model families (ERNIE, Llama, Qwen, Phi). Across this $3 \times 5$ grid, every backbone–LM combination with DSL significantly outperforms all baselines on the **Purity** score by a large margin, demonstrating robust topic assignment accuracy regardless of backbone or SLM choice. The three backbones complement each other across the remaining metrics. Specifically, **ECRTM + DSL** consistently achieves the best topic coherence through $C_V$, **ProdLDA + DSL** achieves the best **LLM** rating, and **FASTopic + DSL** maintains perfect topic diversity (**I-RBO** = 1.000) on all three datasets. The gains are distributed across all four SLM families, indicating that DSL effectively transfers the semantically rich soft targets into the topic model regardless of the SLM teacher's provenance.

Comparing each DSL variant against its corresponding baseline, **ProdLDA + DSL** improves over ProdLDA across every metric on every dataset except for topic diversity, most strikingly on **LLM** rating (e.g., 2.49 → 2.89 on 20News-Group) and **Purity** (e.g., .265 → .803 on StackOverflow). **ECRTM + DSL** similarly improves ECRTM on nearly every column, especially with Purity score (.062 → .805 on StackOverflow). In contrast, **FASTopic + DSL** delivers less consistent gains. While it substantially improves Pu-

---

[3]Hyperparameter details are provided in Appendix D.

[4]Complete results (all $K$) presented in Table 10 of Appendix K.

| Method | 20NewsGroup | | | | TweetTopic | | | | StackOverflow | | | |
|---|---|---|---|---|---|---|---|---|---|---|---|---|
| | $C_V$ | LLM | IRBO | Purity | $C_V$ | LLM | IRBO | Purity | $C_V$ | LLM | IRBO | Purity |
| **Baselines** | | | | | | | | | | | | |
| LDA | .341 | 2.22 | .977 | .301 | .350 | 1.95 | .992 | .441 | .354 | 2.00 | .977 | .174 |
| ProdLDA | .351 | 2.49 | .992 | .356 | .355 | 2.11 | .994 | .533 | .352 | 2.62 | .991 | .265 |
| CombinedTM | .351 | 2.57 | .993 | .391 | .360 | 2.36 | .988 | .588 | .364 | 2.85 | .986 | .306 |
| ZeroShotTM | .354 | 2.53 | .994 | .397 | .356 | 2.30 | .994 | .573 | .363 | 2.84 | .993 | .307 |
| ETM | .344 | 2.35 | .934 | .360 | .351 | 2.22 | .936 | .552 | .348 | 2.28 | .934 | .151 |
| BERTopic | .360 | 2.48 | .992 | .352 | .364 | 2.19 | .996 | .562 | .374 | 2.63 | .996 | .202 |
| ECRTM | .360 | 2.28 | .999 | .364 | .356 | 1.85 | **1.000** | .399 | .373 | 1.95 | **1.000** | .062 |
| FASTopic | .358 | 2.59 | .999 | .416 | .276 | 1.96 | .626 | .557 | .363 | 2.30 | .687 | .171 |
| **ProdLDA + DSL** | | | | | | | | | | | | |
| ERNIE-4.5-0.3B | .381 | 2.86 | .991 | .520 | .392 | 2.90 | .989 | **.781** | .397 | 2.91 | .986 | .737 |
| Qwen-3.5-0.8B | .399 | 2.86 | .980 | .542 | .401 | 2.86 | .976 | **.781** | .403 | 2.89 | .983 | .788 |
| Llama-3.2-1B | .377 | **2.89** | .991 | .564 | .387 | **2.92** | .992 | **.784** | .395 | **2.95** | .991 | .698 |
| Phi-3-mini | .370 | 2.70 | .994 | .557 | .386 | 2.63 | .994 | **.787** | .404 | 2.86 | .988 | **.803** |
| Llama-3.1-8B | .364 | **2.87** | .993 | .559 | .384 | 2.88 | .993 | .774 | .386 | **2.96** | .991 | .700 |
| **FASTopic + DSL** | | | | | | | | | | | | |
| ERNIE-4.5-0.3B | .344 | 2.24 | **1.000** | .510 | .359 | 2.04 | 1.000 | .702 | .384 | 2.27 | **1.000** | .508 |
| Qwen-3.5-0.8B | .347 | 2.15 | 1.000 | .504 | .360 | 1.97 | 1.000 | .695 | .389 | 2.29 | 1.000 | .537 |
| Llama-3.2-1B | .337 | 2.20 | **1.000** | .541 | .357 | 2.03 | 1.000 | .703 | .390 | 2.32 | 1.000 | .530 |
| Phi-3-mini | .336 | 2.04 | 1.000 | .499 | .353 | 1.96 | 1.000 | .705 | .395 | 2.31 | 1.000 | .538 |
| Llama-3.1-8B | .347 | 2.07 | 1.000 | .555 | .355 | 2.01 | 1.000 | .708 | .395 | 2.30 | 1.000 | .542 |
| **ECRTM + DSL** | | | | | | | | | | | | |
| ERNIE-4.5-0.3B | .404 | 2.82 | .985 | .521 | .398 | 2.82 | .983 | .767 | **.410** | 2.85 | .988 | .742 |
| Qwen-3.5-0.8B | **.423** | 2.82 | .975 | .561 | **.406** | 2.82 | .977 | **.781** | .406 | 2.86 | .984 | **.805** |
| Llama-3.2-1B | .404 | 2.84 | .982 | **.582** | .393 | 2.81 | .984 | .777 | .397 | 2.92 | .992 | .708 |
| Phi-3-mini | .375 | 2.78 | .989 | .536 | .383 | 2.72 | .987 | **.778** | .404 | 2.92 | .988 | **.793** |
| Llama-3.1-8B | .388 | 2.83 | .987 | .561 | .386 | 2.82 | .985 | .764 | .393 | **2.95** | .991 | .720 |

*Table 1.* Automatic evaluation results on the top-15 words averaged over 4 numbers of topics ($K = 25, 50, 75, 100$), where results for each $K$ are averaged over 5 random seeds (20 measurements per cell). For each dataset–metric pair, best-performing method and methods that are not significantly different from the best under Welch's independent-samples $t$-test with $\alpha = 0.05$ are **highlighted**.

rity and I-RBO on TweetTopic and StackOverflow where the FASTopic baseline was weakest, on 20NewsGroups it trails the FASTopic baseline on $C_V$ and LLM rating. We attribute this to a target–solver mismatch. FASTopic's Dual Semantic-relation Reconstruction (DSR) OT loss is designed for the *sparse* bag-of-words target, whereas DSL produces a much *denser* top-$k$ soft-label distribution from the SLM. The Sinkhorn solver must then spread topic mass over a wider support, weakening the peakedness that drives coherence metrics. ProdLDA's and ECRTM's direct reconstruction losses absorb the dense target naturally and avoid this trade-off.

From the experiments, **ProdLDA + DSL** offers the best performance–complexity trade-off, where it stays within ∼0.02 of ECRTM + DSL on $C_V$ and Purity across all three datasets while beating ECRTM + DSL on the LLM rating on every dataset, all with the minimal architecture of a vanilla VAE with no embedding clustering regularization, topic-sparsity constraint, or optimal-transport solver. We therefore recommend ProdLDA + DSL as the default for practical applications, even though ECRTM

+ DSL achieves the highest $C_V$ score. It is worth noting that the smallest ERNIE-4.5-0.3B significantly outperforms ZeroShotTM, CombinedTM, BERTopic, and FASTopic, which use an encoder model of a similar size (i.e., GTE-large-en-v1.5) for constructing the contextualized document representation. This demonstrates the benefits of our approach, which allows the topic model to learn high-quality topics through distilling semantic labels from SLMs rather than relying on a sentence-encoder representation alone.

### 4.5. Retrieval Evaluation

While the **Purity** score in Table 1 measures cluster assignment accuracy by evaluating how consistently documents with the same label are grouped into the highest-probable topics, we propose a retrieval-based evaluation to assess the full topic distributions by checking whether documents with similar topic distributions also share the same ground-truth labels. Since all three datasets contain ground-truth labels, we report the label *Precision* for the top-5 (@5) and top-10 (@10) most similar documents according to the KL-

| Method | 20News | | Tweet | | Stack | |
|---|---|---|---|---|---|---|
| | @5 | @10 | @5 | @10 | @5 | @10 |
| **Baselines** | | | | | | |
| LDA | .244 | .231 | .674 | .500 | .113 | .109 |
| ProdLDA | .346 | .334 | .767 | .619 | .242 | .231 |
| ZeroShotTM | .384 | .373 | .776 | .647 | .280 | .268 |
| CombinedTM | .366 | .357 | .744 | .637 | .260 | .251 |
| ETM | .312 | .305 | .786 | .637 | .135 | .125 |
| BERTopic | .413 | .402 | .546 | .519 | .306 | .296 |
| ECRTM | .341 | .329 | .712 | .530 | .114 | .101 |
| FASTopic | .421 | .406 | .811 | .667 | .189 | .178 |
| **ProdLDA + DSL** | | | | | | |
| ERNIE-4.5-0.3B | .509 | .499 | .900 | .844 | .754 | .746 |
| Llama-3.1-8B | .577 | .565 | .909 | .854 | .784 | .772 |
| Llama-3.2-1B | .553 | .543 | .904 | .846 | .752 | .741 |
| Qwen-3.5-0.8B | .516 | .508 | .902 | .847 | .805 | .799 |
| Phi-3-mini | .563 | .553 | .900 | .847 | .834 | .827 |
| **ECRTM + DSL** | | | | | | |
| ERNIE-4.5-0.3B | .559 | .542 | .924 | .859 | .799 | .789 |
| Llama-3.1-8B | **.634** | **.614** | **.935** | **.877** | .842 | .829 |
| Llama-3.2-1B | .594 | .579 | .925 | .862 | .800 | .788 |
| Qwen-3.5-0.8B | .554 | .542 | .925 | .861 | .834 | .827 |
| Phi-3-mini | .624 | .603 | .933 | **.877** | **.867** | **.860** |
| **FASTopic + DSL** | | | | | | |
| ERNIE-4.5-0.3B | .409 | .407 | .616 | .611 | .395 | .391 |
| Llama-3.1-8B | .459 | .458 | .627 | .622 | .416 | .414 |
| Llama-3.2-1B | .445 | .444 | .612 | .609 | .409 | .406 |
| Qwen-3.5-0.8B | .410 | .408 | .614 | .605 | .421 | .418 |
| Phi-3-mini | .415 | .412 | .626 | .616 | .453 | .447 |

*Table 2.* Retrieval evaluation results for precision @5 and @10, averaged over four numbers of topics ($K = 25, 50, 75, 100$) × five random seeds (20 measurements per cell). For each dataset–metric pair, best-performing method and methods that are not significantly different from the best under Welch's independent-samples $t$-test with $\alpha = 0.05$ are **highlighted**.

divergence of the predicted topic distributions in Table 2.

From the results, we see that **ProdLDA + DSL** and **ECRTM + DSL** significantly outperform all baselines by a wide margin, with **ECRTM + DSL** achieving the strongest retrieval performance overall. The largest gains appear on StackOverflow, where the best DSL variant nearly triples the best baseline (.306 → .867 P@5), demonstrating the superiority of our proposal in handling data sparsity for short-text retrieval. The one exception is **FASTopic + DSL**, which trails the other two DSL backbones on every dataset and even underperforms the FASTopic baseline on Tweet-Topic (.811 → .627 P@5). This is consistent with our earlier finding that FASTopic's optimal-transport objective over-diversifies topics under dense SLM targets (I-RBO = 1.000), weakening the fine-grained topic-overlap signal that retrieval requires. Lastly, our low-dimensional topic representations offer substantial computational and storage advantages over the high-dimensional dense embeddings produced by pre-trained encoder models. We hope this find-

ing will inspire future works on better leveraging SLMs to learn topics for retrieval-oriented applications.

## 5. Analysis

### 5.1. Ablation Studies

To assess the individual contributions of each innovation in our proposed method, we conduct ablation experiments with ProdLDA to examine the effects of the three key components described in §3, namely, the reconstruction target, the training loss, and the topic model input. We propose the following four ablations: (1) Rather than the temperature-scaled KL-Divergence as the reconstruction loss, we instead use the standard negative log-likelihood from ProdLDA (**NLL**). (2) Using the NLL loss, we further replace the soft label targets with the BoW document representation as the reconstruction target (**NLL + BoW**). (3) The final hidden state from language model is replaced with the contextualized embeddings from the similar sized GTE-large-en-v1.5 (**Embedding**). (4) We use GTE-large-en-v1.5 embeddings as input along with the BoW reconstruction target and NLL loss, this is equivalent to ZeroShotTM (Bianchi et al., 2021b) (**NLL + BoW + Embedding**).

All the ablation study results are presented in Table 3. Our first observation is that replacing the KL loss with the standard NLL results in higher $C_V$, but lower topic diversity and purity, representing a trade-off between the objectives. Since we use KL with temperature scaling, a temperature value $\tau > 1$ flattens the distribution, resulting a higher utilization of lower-probability words. This encourages the topics to cover capture different parts of the vocabulary, leading to more diverse but slightly less concentrated topics (we show the effects of temperature $\tau$ in subsection 5.2). More importantly, we find that using the standard BoW reconstruction target (NLL + BoW) leads to the largest drop in performance, demonstrating that our semantically-grounded soft targets most significantly contributes to the topic quality and assignment accuracy. Lastly, we find that replacing the SLM hidden states with GTE-large-en-v1.5 embeddings (Embedding) leads to a slight drop in performance compared to original, where using the general purpose GTE-large-en-v1.5 embeddings to reconstruct the BoW objective (NLL + BoW + Embedding) can even lead to a small increase in topic coherence. This shows that using the SLM's last hidden states is only beneficial for reconstructing our soft label targets and does not offer a significant advantage in the standard topic modeling objective.

| Ablation | 20NewsGroup | | | | TweetTopic | | | | StackOverflow | | | |
|---|---|---|---|---|---|---|---|---|---|---|---|---|
| | $C_V$ | LLM | IRBO | Purity | $C_V$ | LLM | IRBO | Purity | $C_V$ | LLM | IRBO | Purity |
| **ERNIE-4.5-0.3B-PT** | | | | | | | | | | | | |
| Original | .381 | 2.86 | .991 | .520 | .392 | 2.90 | .989 | .781 | .397 | 2.91 | .986 | .737 |
| NLL | **+.024** | -.09 | -.023 | -.046 | **+.014** | -.13 | -.024 | -.019 | **+.022** | -.08 | -.011 | -.021 |
| NLL + BoW | -.042 | -.38 | +.000 | -.088 | -.034 | -.73 | **+.005** | -.157 | -.028 | -.15 | **+.007** | -.363 |
| Embedding | +.000 | -.01 | -.003 | -.007 | **+.005** | -.02 | -.004 | -.060 | -.001 | -.01 | -.007 | -.224 |
| NLL + BoW + Embedding | -.027 | -.33 | **+.003** | -.123 | -.035 | -.59 | **+.004** | -.208 | -.034 | -.07 | **+.007** | -.430 |
| **Llama-3.2-1B-Instruct** | | | | | | | | | | | | |
| Original | .377 | 2.89 | .991 | .564 | .387 | 2.92 | .992 | .784 | .395 | 2.95 | .991 | .698 |
| NLL | **+.021** | -.15 | -.023 | -.061 | **+.010** | -.06 | -.018 | -.018 | **+.004** | -.11 | -.012 | -.049 |
| NLL + BoW | -.036 | -.42 | +.000 | -.121 | -.033 | -.78 | **+.004** | -.175 | -.032 | -.19 | **+.002** | -.363 |
| Embedding | **+.005** | -.00 | -.002 | -.016 | **+.003** | -.01 | -.003 | -.061 | **+.008** | -.04 | -.006 | -.230 |
| NLL + BoW + Embedding | -.023 | -.36 | **+.003** | -.167 | -.030 | -.62 | **+.002** | -.211 | -.032 | -.11 | +.002 | -.391 |
| **Llama-3.1-8B-Instruct** | | | | | | | | | | | | |
| Original | .364 | 2.87 | .993 | .559 | .384 | 2.88 | .993 | .774 | .386 | 2.96 | .991 | .700 |
| NLL | **+.032** | -.11 | -.020 | -.059 | **+.006** | **+.03** | -.035 | -.005 | **+.011** | -.04 | -.012 | +.001 |
| NLL + BoW | -.026 | -.40 | -.001 | -.107 | -.026 | -.76 | **+.002** | -.169 | -.023 | -.22 | +.001 | -.354 |
| Embedding | +.002 | -.02 | -.003 | -.007 | -.001 | -.02 | -.003 | -.058 | -.002 | -.014 | -.006 | -.218 |
| NLL + BoW + Embedding | -.010 | -.34 | **+.001** | -.162 | -.027 | -.58 | **+.001** | -.201 | -.023 | -.12 | +.001 | -.393 |

*Table 3.* Ablation experiment results with our three models ERNIE-4.5-0.3B, Llama-3.2-1B, and Llama-3.1-8B averaged over $K = 25, 50, 75, 100$, with five random seeds per $K$. Results that are statistically significantly better and worse than the Original ($p < 0.05$) are highlighted in green and red, respectively.

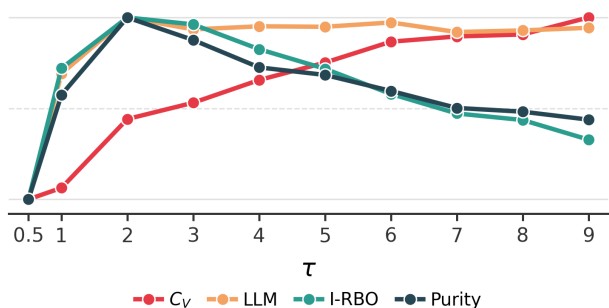

*Figure 3.* Results on 20NewsGroups with ERNIE-4.5-0.3B for various temperature $\tau$ values. The values are normalized to visualize relative trends.

## 5.2. Temperature Values

To investigate the effects of temperature $\tau$ on the topic qualities, we conduct experiments with various $\tau$ values and visualize the results for 20NewsGroups with ERNIE-4.5-0.3B in Figure 3[5]. Since the softmax temperature $\tau$ (Equation 1) controls the sharpness of the distribution, small values (i.e., $\tau < 2$) cause the softmax to approach an argmax operation. This concentrates gradients on only a few vocabulary words during reconstruction, leading the topic model to lose semantic breadth, resulting in learned topics are neither coherent nor diverse. In

---

[5]Similar trends on the two other datasets are visualized in Figure 4 of Appendix E.

contrast, at higher temperatures ($\tau > 5$), the model learns to reconstruct a softer distribution in which thematically similar words are more likely to co-occur within the same topic, yielding higher coherence but reduced diversity. Accordingly, we choose $\tau = 3$ as a balance between semantic informativeness and redundancy.

## 5.3. Instruction Prompt Sensitivity

To quantify whether topic model performance varies with the instruction prompt, we conduct experiments on the 20NewsGroups dataset with ERNIE-4.5-0.3B for $K = 50$ using 5 different prompts[6]. For each prompt, we train 5 random seeds and apply the two-way Analysis of Variance (ANOVA) test (Scheffe, 1999) to compare variance between prompt choice and random seed. For coherence metrics, the estimated main effects of prompt and seed are statistically indistinguishable for $\mathbf{C_V}$ ($F_{prompt} = 1$, $F_{seed} = 1$) and **LLM** ratings ($F_{prompt} = 0.72$, $F_{seed} = 0.37$), indicating that changing the prompt wording has no more effect than simply re-running with a different random seed. For topic diversity and **Purity**, prompt choice explains approximately 10 times more variance than seed choice for **I-RBO** ($F_{prompt} = 42.4$, $F_{seed} = 4.4$) and 5 times more for **Purity** ($F_{prompt} = 9.8$, $F_{seed} = 2.1$). This is likely due to the instruction steering the language model towards a particular facet, therefore changing the breadth of topic coverage.

---

[6]The five prompt variants are presented in Appendix F.

| Method | Top-15 Topic Words (*sci.med* – 20NewsGroup, $K = 75$) |
|---|---|
| LDA (.325) | pain , problem , doctor , effect , fat , reaction, recall , extremely , like , happen , pictures, ago , paper , sorry , think |
| ProdLDA (.467) | oil , water , battery , pain, temperature , cause , food , reaction , weight , heat , cold , air , pressure , brain, effects |
| ZeroShotTM (.815) | doctor , pain , medical , patient , symptoms , treatment , doctors , medicine , tests , diet , patients , literature, sick, day, vitamin |
| CombinedTM (.497) | water , brain, food , blood , fat , heart, cause , energy , doctor , effects , heat , pain , reaction, eat , levels |
| ETM (.342) | new , problem , high , end, large , cost , found, long , level , comes , numbers , number , problems , price , year |
| BERTopic (.821) | patients , disease , health , doctor , medical , cancer , vitamin , treatment , infection, patient , diet , pain , doctors , insurance, food |
| ECRTM (.705) | doctor , vitamin , blood , cancer , brain, describes, patients , causes , infection , symptoms , disease , age , gain, diet , women |
| FASTopic (.356) | easier , comparison, differences , requirements , usually , reduce , compare , fairly , slightly , limit , easily, advantage , require , equivalent , depends |
| **ProdLDA + DSL** (.877) | clinical , doctor , medical , disease , treatment , patient , symptoms , doctors , infection , pain, body, medicine , care , patients , cancer |

| Method | Top-15 Topic Words (*talk.politics.misc* – 20NewsGroup, $K = 75$) |
|---|---|
| LDA (.540) | people , children , know , said , war , time, like, world , life , country , come , right , man , think , way |
| ProdLDA (.526) | figures , study , crimes , criminal , men , gun , crime , statistics , gay, kill , sexual , firearms , published, violent , guns |
| ZeroShotTM (.734) | insurance , tax , private, taxes , costs , doctors, pay , paying , spend , health , spending , paid , money , price, market |
| CombinedTM (.690) | men , gay , male , straight, homosexuals , sexual , homosexual , sex , article, study , women , report, percent , behavior , homosexuality |
| ETM (.523) | know , people , think , time, way , good , things , want , years, course , day, come , point , going , thing |
| BERTopic (.579) | government , taxes , tax , party, billion , dollars , economy , pay , spending , power , money , parties, state , people, society |
| ECRTM (.734) | homosexuality , sexual , verses , sin , passage , teaching, ancient , quotes, men , sins , relationship, behavior , homosexual , pray , desire |
| FASTopic (.695) | entry , file , code , program , include , output , section , write , char , files , build, return, int , function, line |
| **ProdLDA + DSL** (.778) | sexual , homosexuals , homosexual , homosexuality , sex , gay , male , marriage , women, men , relationship , society , social , woman , family |

*Table 4.* Top-15 words of each method's most aligned topic with the two ground-truth classes (*sci.med* and *talk.politics.misc*), based on the inverse Purity score. Each word's background shading encodes its leave-one-out $C_V$ contribution within its topic (darker red = higher contribution; unshaded = negligible contribution). The per-topic $C_V$ score is shown in parentheses after each method name.

## 5.4. Qualitative Results

We provide qualitative examples in Table 4 to inspect the topics produced by each method on the 20NewsGroups dataset. For each method we display the topic most aligned with a given ground-truth class, where alignment is computed as the topic containing the largest number of documents from that class (equivalently, the topic with the highest inverse Purity score). We select two ground-truth classes from **ProdLDA + DSL** with `ERNIE-4.5-0.3B` on per-topic $C_V$ ( *sci.med* and *talk.politics.misc*), both at $K = 75$. Within each topic, the per-word highlight color encodes the respective leave-one-out $C_V$ contribution, where darker red words anchor the topic's coherence, while gray words are only weakly bonded to the rest of the cluster.

Our method captures the core terminology of each class. For *sci.med*, ProdLDA + DSL produces a tightly clinical cluster anchored on "clinical", "patient", "patients", and "treatment", achieving a per-topic $C_V$ of .877 versus the strongest baseline ceiling of .821 (BERTopic). For *talk.politics.misc*, our model surfaces a coherent civil-rights debate cluster anchored on "sexual", "homosexuality", and "sex", again with the highest per-topic $C_V$ (.778). The peripheral words in both rows ("cancer", "body", "care" for medicine; "society", "social", "family" for politics) stay on-theme even when their individual LOO contribution is low, indicating that the dense soft-label target draws in semantically related vocabulary that may not appear in any single document.

## 6. Conclusion

We propose a novel technique for enhancing neural topic modeling that uses the NLU capabilities of the language models to generate a soft label distribution as the reconstruction target for neural topic models. This is done by projecting the temperature-scaled softmax distribution of the next token prediction immediately following the prompt onto our defined vocabulary, before training the neural topic model using the final hidden state as input. In extensive experiments on three datasets, we demonstrated that our approach achieves state-of-the-art performance in terms of topic coherence and purity, while maintaining high diversity. We also introduce a retrieval-based evaluation metric where we showed that the predicted topic distribution from our method achieved the highest precision for retrieving semantically similar documents according to the ground-truth label.

In future work, we will experiment with different prompting strategies such as few-shot learning (Brown et al., 2020) and chain-of-thought (CoT) prompting (Wei et al., 2022) to more effectively generate the soft label targets. We will also adapt our approach to perform multi-modal topic modeling using different types of vision-language models. Lastly, we will also explore the effectiveness of using the inferred topic for retrieval-based downstream applications, which has proven to be a promising direction based on the results from our experiments.

## Acknowledgments

The authors thank the anonymous reviewers and the Area Chair for their valuable feedback and suggestions. The authors acknowledge the support of the Natural Sciences and Engineering Research Council of Canada (NSERC). Nous remercions le Conseil de recherches en sciences naturelles et en génie du Canada (CRSNG) de son soutien.

## Impact Statement

This paper presents work with goal of improving neural topic modeling through semantically enriched supervision from small language models. Topic models are commonly used to support exploration, organization, retrieval, and summarization of large text corpora. By improving topic coherence, assignment accuracy, and retrieval-oriented document representations, our method may help researchers and practitioners analyze large collections of text more effectively while preserving the interpretable document-topic and topic-word structure of probabilistic topic models.

The main societal risks of this work arise from the data and language models used in topic modeling applications. Real-world text corpora may contain private information, harmful stereotypes, misinformation, or other biased and inaccurate content. Since our method distills semantic signals from language models, it may also inherit or amplify biases encoded in those models, leading to topics or document assignments that appear coherent but reflect undesirable social assumptions. In sensitive domains, such as healthcare, hiring, education, political communication, or user profiling, topic models should therefore be treated as exploratory tools rather than definitive decision-making systems.

There is also a potential risk that improved topic modeling and retrieval could be used for undesirable forms of large-scale monitoring, profiling, or manipulation of individuals or communities. These risks are not unique to our method, but stronger semantic topic representations may make such applications easier. Potential mitigations include applying appropriate privacy protections and data governance procedures, filtering or auditing vocabularies and corpora before deployment, evaluating topic quality across demographic and domain-specific subgroups where relevant, and involving domain experts when interpreting topics in high-stakes settings.

Our experiments were conducted on standard public benchmark datasets in controlled research settings. We do not deploy the proposed method in any real-world decision-making system. We encourage future applications of this work to report dataset provenance, pre-processing choices, vocabulary filtering criteria, model-selection decisions, and known limitations, especially when topic models are used to analyze sensitive or user-generated text.

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

## A. Datasets

| Dataset | Domain | # Docs | # Labels | Avg. Words |
|---|---|---|---|---|
| 20NewsGroups | Email | 18,846 | 20 | 182 |
| TweetTopic | Tweet | 21,996 | 19 | 28 |
| StackOverflow | Forum | 20,000 | 20 | 9 |

*Table 5.* Statistics of the three datasets used in our experiments.

Table 5 summarizes the datasets used in section 4. We evaluate on three publicly available text classification corpora with ground-truth labels: 20NewsGroup[7], TweetTopic[8] (Antypas et al., 2022), and StackOverflow[9] (Xu et al., 2015). For TweetTopic, we keep only single-label examples so that Purity and retrieval precision can be computed against one reference label per document. TweetTopic and StackOverflow are short-text datasets, where sparse BoW reconstruction is especially challenging.

## B. Evaluation Metrics

**Topic Coherence** For automatic coherence, we use $C_V$ (Röder et al., 2015), which has been shown to outperform earlier co-occurrence metrics such as NPMI, UCI, and UMass in correlating with human judgments (Newman et al., 2010; Stevens et al., 2012; Lau et al., 2014; Röder et al., 2015). Following prior work (Röder et al., 2015; Wu et al., 2023; 2024a), we compute word co-occurrences using a

[7] http://qwone.com/~jason/20Newsgroups
[8] https://huggingface.co/datasets/cardiffnlp/tweet_topic_multi
[9] https://github.com/jacoxu/StackOverflow

large Wikipedia reference corpus through the Palmetto[10] toolkit.

We also report **LLM** ratings following Stammbach et al. (2023). Specifically, we use gpt-4o through the OpenAI API[11] to score the top words of each topic on a 1–3 relatedness scale. The system prompt is shown in Table 6.

---
**LLM Rating System Prompt**

You are a helpful assistant evaluating the top words of a topic model output for a given topic. Please rate how related the following words are to each other on a scale from 1 to 3 ("1" = not very related, "2" = moderately related, "3" = very related).

Reply with a single number, indicating the overall appropriateness of the topic.

---
*Table 6.* System prompt used for the **LLM** topic-coherence rating.

**Topic Diversity** We use Inverse Rank-Biased Overlap (**I-RBO**) (Terragni et al., 2021b; Bianchi et al., 2021a). Unlike standard Topic Diversity (Dieng et al., 2020), which only measures set overlap, I-RBO is rank-aware and penalizes overlap more strongly among top-ranked words that dominate human interpretation.

**Topic Assignment Accuracy** Following prior studies (Poursabzi-Sangdeh et al., 2016; Hoyle et al., 2022), we evaluate assignment accuracy with harmonic **Purity** (Zhao & Karypis, 2001). Each document is assigned to its highest-probability topic, and each ground-truth class is matched to the topic that maximizes the F1 score. The final score is aggregated with class-size weighting.

**Retrieval Precision** To evaluate the full document-topic distribution rather than only the top-1 topic assignment, we retrieve documents using the KL divergence between predicted topic distributions. For a query document $i$, let $R_N(i)$ denote the $N$ nearest documents according to $D_{KL}(\theta_i\|\theta_j)$ (excluding query itself), where $\ell_i$ is the ground-truth label of document $i$ (Equation 6).

$$\text{P@}N = \frac{1}{|\mathcal{D}|}\sum_{i\in\mathcal{D}}\frac{1}{N}\sum_{j\in R_N(i)}\mathbf{1}\{\ell_i=\ell_j\} \qquad (6)$$

## C. Baselines

We compare against eight baselines spanning LDA-family neural topic models, clustering-based topic models, and recent regularized neural topic models. For LDA, ProdLDA, CombinedTM, ZeroShotTM, and ETM, we use the OCTIS library (Terragni et al., 2021a) with the default settings for each model unless otherwise specified.

[10] https://github.com/dice-group/Palmetto
[11] https://openai.com/api

**LDA** We use the `Gensim` implementation (Řehůřek & Sojka, 2010) provided by OCTIS. The model uses online variational Bayes (Hoffman et al., 2010) with a batch size of 2000, one pass through the corpus, a gamma convergence threshold of 0.001, learning-rate decay 0.5, and offset 1.0.

**ProdLDA** We use the OCTIS implementation of ProdLDA (Srivastava & Sutton, 2017), which employs a VAE with a product-of-experts decoder. The encoder has two hidden layers with 200 neurons, softplus activations (Zheng et al., 2015), and dropout rate 0.2. The model is trained with Adam (Kingma & Ba, 2015), learning rate $2 \times 10^{-3}$, batch size 64, and 100 epochs. Following the default setup, the prior parameters are optimized and inference averages 10 samples.

**CombinedTM and ZeroShotTM** CombinedTM (Bianchi et al., 2021a) uses the same VAE architecture as ProdLDA but concatenates the BoW input with contextualized sentence embeddings. ZeroShotTM (Bianchi et al., 2021b) uses contextualized embeddings as input without the BoW vector. In our experiments, the contextualized embeddings are produced by `GTE-large-en-v1.5` rather than SBERT, as described in Appendix D.

**ETM** The Embedding Topic Model (Dieng et al., 2020) represents topics and words in a shared embedding space. We initialize word embeddings with `word2vec-google-news-300` (Mikolov et al., 2013) downloaded through `Gensim`; these embeddings are not fine-tuned. The encoder maps normalized BoW vectors to topic proportions through a two-layer network with hidden size 800, ReLU activation, dropout rate 0.5, Adam learning rate $5 \times 10^{-3}$, weight decay $1.2 \times 10^{-6}$, and early stopping with patience 5.

**BERTopic** We use the official BERTopic implementation[12] (Grootendorst, 2022). Documents are embedded with the same contextual encoder used for the embedding-based baselines, projected to 5 dimensions using UMAP (McInnes et al., 2018), and clustered with HDBSCAN (Campello et al., 2013). Topic-word distributions are computed using class-based TF-IDF over the documents assigned to each cluster. HDBSCAN outliers are excluded from topic evaluation.

**ECRTM** We adopt the original ECRTM implementation[13] (Wu et al., 2023). ECRTM augments a VAE topic model with embedding clustering regularization to separate topic embeddings and mitigate topic collapse. The encoder has two hidden layers with 200 neurons and softplus activa-

tions. Topic and word embeddings are 200-dimensional; the topic-word matrix is computed with a softmax over negative pairwise Euclidean distances using temperature 0.2. Word embeddings are initialized with 200-dimensional GloVe embeddings (Pennington et al., 2014). The ECR term is solved as an optimal transport problem with Sinkhorn iterations (Sinkhorn, 1964; Cuturi, 2013), using $\alpha = 20.0$, at most 1000 iterations, and loss weight $\lambda_{\mathrm{ECR}} = 100.0$. The model is trained for 200 epochs with Adam, learning rate 0.002, and batch size 64.

**FASTopic** We use the TopMost implementation (Wu et al., 2024b) of FASTopic (Wu et al., 2024a). FASTopic encodes documents with contextualized embeddings and uses optimal-transport regularization to align documents, topics, and words while retaining BoW reconstruction. We use the default hyperparameters: document-topic regularization $\alpha_{DT} = 3.0$, topic-word regularization $\alpha_{TW} = 2.0$, learning rate 0.002, and 200 training epochs.

## D. Hyperparameter Settings

For all datasets, we build the topic-modeling vocabulary $V$ from the top-$|V|$ most frequent words after tokenization and stop-word removal, and use $|V| = 2000$ in the main experiments. As described in section 4, we keep vocabulary words that correspond to single tokens under the LM tokenizer for efficient next-token projection.

For **ProdLDA + DSL**, we use the same architecture and optimizer settings as the ProdLDA baseline: a two-layer MLP encoder with hidden size 200 and softplus activation, Adam with learning rate $2 \times 10^{-3}$ and cosine decay, batch size 64, and 100 epochs. For **ECRTM + DSL** and **FASTopic + DSL**, we keep the original model-specific architecture and regularization settings described in Appendix C; DSL only replaces the BoW reconstruction target with the LM-induced target $y_{\mathrm{DSL}}$ and uses the KL reconstruction term in Equation 3.

For DSL-specific hyperparameters, we use temperature $\tau = 3$ in Equation 1 and reconstruction weight $\lambda = 1e^3$ in Equation 3. For baselines requiring contextualized document embeddings, namely CombinedTM, ZeroShotTM, BERTopic, and FASTopic, we use `GTE-large-en-v1.5`[14] (Zhang et al., 2024), a 0.4B dense encoder with hidden size 1024.

## E. Temperature Analysis

The experiment results for different temperature values $\tau$ on 20NewsGroup, TweetTopic, and StackOverflow are visualized in Figure 4. Across datasets, we observe similar

---

[12]https://github.com/MaartenGr/BERTopic
[13]https://github.com/BobXWu/ECRTM

[14]https://huggingface.co/Alibaba-NLP/gte-large-en-v1.5

qualitative trends. Topic diversity and assignment accuracy tend to peak near $\tau \approx 3$, while very small values make the softmax target close to a one-hot distribution and very large values make the target overly diffuse. StackOverflow shows a weaker diversity drop at high temperature, likely because its short technical titles induce more varied LM targets. These trends support the default choice $\tau = 3$ used in the main experiments.

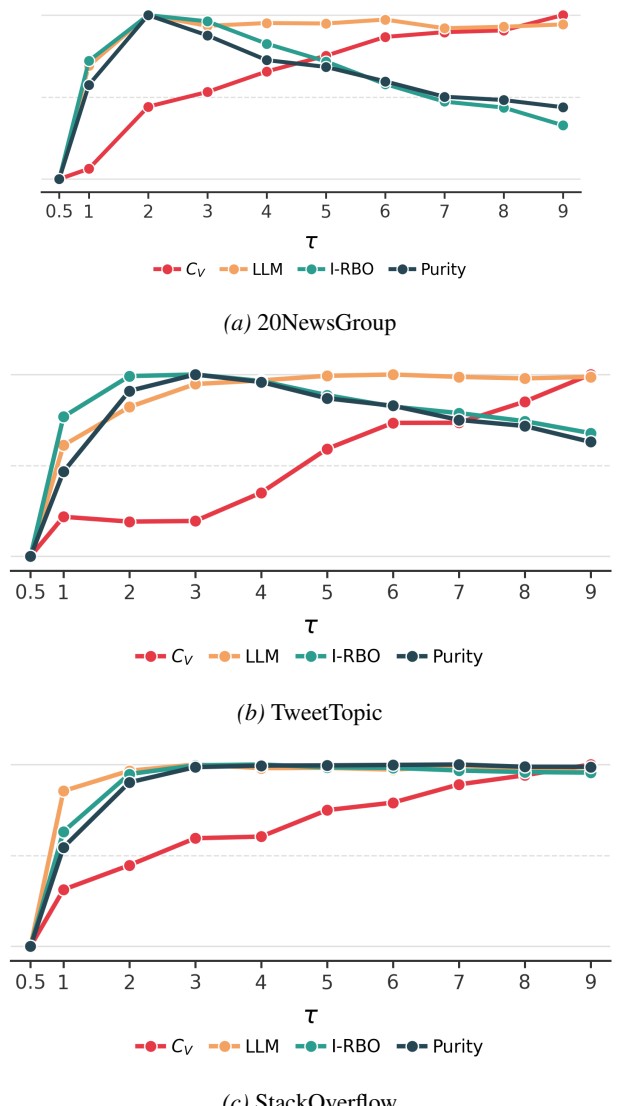

*(a) 20NewsGroup*

*(b) TweetTopic*

*(c) StackOverflow*

*Figure 4.* Results on 20NewsGroup, TweetTopic, and StackOverflow with `ERNIE-4.5-0.3B` for different temperature values $\tau$. Metric values are normalized to visualize relative trends.

## F. Prompts for Sensitivity Analysis

Using `gpt-4o`, we generated five re-phrasings of the instruction prompt, "Generate a single-word label that captures the overall theme of the document below." The resulting prompt variants are shown in Table 7. These are the

prompts used for the prompt-sensitivity analysis in the main text.

| **Instruction Prompt Variants** |
|---|
| 1. *Name the document's central theme.* |
| 2. *Give a single label that sums up the text.* |
| 3. *Tag the document with its main idea.* |
| 4. *Read the following document in full, identify its principal theme, and output a one-word label that best conveys that unifying concept.* |
| 5. *Generate a tag that captures the primary theme running through the entire document.* |

*Table 7.* Five rephrased instruction prompts used to construct soft labels for the prompt-sensitivity analysis.

## G. Experiments with Different Vocabulary Sizes

We present results with different vocabulary sizes in Figure 5. DSL with `ERNIE-4.5-0.3B` achieves the best or near-best performance across vocabulary sizes. The performance gap between DSL and BoW-based baselines generally increases as vocabulary size grows, which is consistent with our motivation: dense semantic targets can assign useful mass to theme-relevant words beyond the observed document tokens, while sparse BoW targets become increasingly incomplete as the candidate vocabulary expands.

## H. Layer Selection for LM Hidden-State Inputs

Table 8 evaluates which representation should be used as the topic-model input as different layers could encode different features (Niu et al., 2022; Jin et al., 2025). We find that the final hidden state achieves the best overall performance as they are directly used by the SLM head to predict the next-token logits, with a slight performance drop at Layer 12 and a larger drop at Layer 6. This supports our design choice in subsection 3.3.

| Input Representation | $C_V$ | LLM | I-RBO | Purity |
|---|---|---|---|---|
| `GTE-large-en-v1.5` | .381 | 2.85 | .988 | .513 |
| Layer 6 (middle) | .379 | 2.68 | .982 | .501 |
| Layer 12 (3/4 depth) | .377 | 2.83 | .991 | .512 |
| Final layer (ours) | .381 | 2.86 | .991 | .520 |

*Table 8.* Layer-selection experiment on 20NewsGroups using `ERNIE-4.5-0.3B`. Results are averaged across five number of topics ($K = 25, 50, 75, 100$), where each $K$ is averaged across five random seeds.

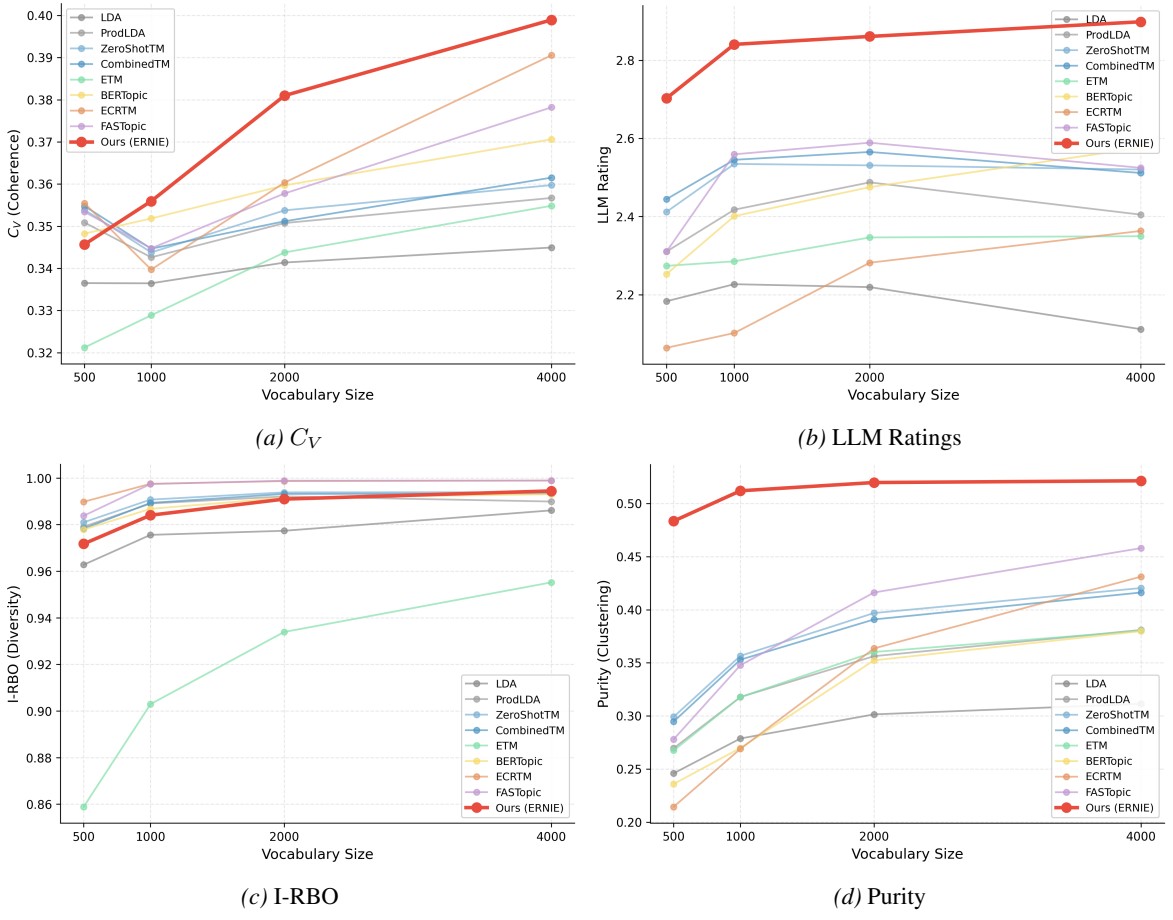

*Figure 5.* Effect of vocabulary size on topic model performance across four evaluation metrics on 20NewsGroup. Results are shown for vocabulary sizes 500, 1,000, 2,000, and 4,000. DSL with `ERNIE-4.5-0.3B` is shown in red.

## I. Target Sparsity and Topic Diversity

We run a small-scale experiment to check whether topic diversity can be controlled by sparsifying the dense target distribution. In Table 9, we keep only the top-$k$ logits of $y_{\text{DSL}}$ before renormalization on TweetTopic. Sparse DSL targets improve I-RBO relative to the full target while preserving large gains in LLM rating and Purity over BoW-based baselines, confirming that the coherence-diversity trade-off can be tuned through target sparsity.

| Method | Top Words | $C_V$ | LLM | I-RBO | Purity |
|---|---|---|---|---|---|
| ProdLDA | $\sim$5.2 (BoW) | .352 | 2.14 | .994 | .532 |
| ZeroShotTM | $\sim$5.2 (BoW) | .351 | 2.31 | .994 | .575 |
| DSL | 10 | .388 | 2.66 | .998 | .744 |
| DSL | 50 | .384 | 2.79 | .998 | .771 |
| DSL | 100 | .380 | 2.79 | .997 | .771 |
| DSL | 500 | .383 | 2.92 | .996 | .779 |
| DSL | 1000 | .383 | 2.90 | .992 | .780 |
| DSL | Full | .388 | 2.88 | .989 | .779 |

*Table 9.* Target-sparsity analysis on TweetTopic using `ERNIE-4.5-0.3B`. Top-$k$ DSL targets are re-normalized after retaining the $k$ largest entries. Results are computed for $K = 50$.

## J. Mathematical Justification of Semantic Reconstruction

### J.1. Implicit Posterior Predictive Target

Let $V$ denote the fixed topic-modeling vocabulary. For each document $x$ and thematic instruction prompt $\pi$, the frozen LM produces next-token logits $\ell_{\text{LM}}(x, \pi)$. Restricting these logits to $V$ gives $\ell_V(x, \pi) \in \mathbb{R}^{|V|}$, and the DSL target is

$$y_{\text{DSL}}(x, \pi) = \text{softmax}\left( \frac{\ell_V(x, \pi)}{\tau} \right) \in \Delta^{|V|-1}. \quad (7)$$

Following the implicit Bayesian interpretation of language models, the prompt-conditioned next-token distribution can be viewed as a posterior predictive distribution over words after marginalizing an implicit latent concept $c$:

$$p_{\text{LM}}(v \mid x, \pi) \approx \int p_{\text{LM}}(v \mid c)\, p_{\text{LM}}(c \mid x, \pi)\, dc. \quad (8)$$

The target $y_{\text{DSL}}$ is the vocabulary-restricted and temperature-scaled version of this semantic posterior predictive signal. We do not assume that the latent concept

$c$ is observed or explicitly represented by the LM; it is used only to motivate why the next-token distribution contains thematic information beyond the observed BoW support.

### J.2. Structured Topic-Model Family

For a topic model $\mathcal{M}$ with trainable parameters $\psi$, let $\hat{y}_\psi(\cdot \mid x_{\mathrm{emb}})$ denote the vocabulary-level predictive distribution produced for document representation $x_{\mathrm{emb}}$. The set of distributions representable by the model for a fixed document is

$$\mathcal{P}_\mathcal{M}(x_{\mathrm{emb}}) = \{\hat{y}_\psi(\cdot \mid x_{\mathrm{emb}}) : \psi \in \Psi_\mathcal{M}\} \subseteq \Delta^{|V|-1}. \quad (9)$$

For VAE-based topic models such as ProdLDA, the encoder defines a variational posterior $q_\phi(z \mid x_{\mathrm{emb}})$ over a $K$-dimensional latent topic variable, with document-topic proportions $\theta = \mathrm{softmax}(z)$. The decoder and topic-word parameters $\boldsymbol{\beta}$ induce

$$\hat{y}_{\phi,\boldsymbol{\beta}}(\cdot \mid x_{\mathrm{emb}}) = \mathbb{E}_{z \sim q_\phi(z|x_{\mathrm{emb}})}\left[p(\cdot \mid z, \boldsymbol{\beta})\right]. \quad (10)$$

Thus, unlike the unstructured LM next-token distribution, every distribution in $\mathcal{P}_\mathcal{M}(x_{\mathrm{emb}})$ must pass through the topic bottleneck and is tied to corpus-level topic-word parameters.

### J.3. Forward-KL Reconstruction as Dense Semantic Cross-Entropy

The DSL reconstruction term in Equation 3 is

$$D_{\mathrm{KL}}(y_{\mathrm{DSL}}(x, \pi) \,\|\, \hat{y}_\psi(\cdot \mid x_{\mathrm{emb}})) \quad (11)$$

$$= \sum_{v \in V} y_{\mathrm{DSL}}(v \mid x, \pi) \log \frac{y_{\mathrm{DSL}}(v \mid x, \pi)}{\hat{y}_\psi(v \mid x_{\mathrm{emb}})}. \quad (12)$$

Since the entropy of $y_{\mathrm{DSL}}$ is fixed w.r.t. $\psi$, minimizing Equation 12 is equivalent to minimizing the CE:

$$-\sum_{v \in V} y_{\mathrm{DSL}}(v \mid x, \pi) \log \hat{y}_\psi(v \mid x_{\mathrm{emb}}). \quad (13)$$

Therefore, DSL is the direct analogue of BoW reconstruction with a dense, LM-induced semantic target replacing the sparse empirical word-count target.

**Exact Realizability**  If $y_{\mathrm{DSL}}(x, \pi) \in \mathcal{P}_\mathcal{M}(x_{\mathrm{emb}})$, then there exists $\psi^\star$ such that $\hat{y}_{\psi^\star}(\cdot \mid x_{\mathrm{emb}}) = y_{\mathrm{DSL}}(x, \pi)$, and the reconstruction KL is zero.

**Approximate Realizability**  If $y_{\mathrm{DSL}}(x, \pi) \notin \mathcal{P}_\mathcal{M}(x_{\mathrm{emb}})$, then minimizing the reconstruction term yields

$$\hat{y}_{\psi^\star}(\cdot \mid x_{\mathrm{emb}}) \in \arg\min_{p \in \mathcal{P}_\mathcal{M}(x_{\mathrm{emb}})} D_{\mathrm{KL}}(y_{\mathrm{DSL}}(x, \pi) \,\|\, p), \quad (14)$$

which is the closest structured topic-model approximation to the LM-induced semantic target under the forward KL direction used in the main objective.

### J.4. Relation to Bag-of-Words Reconstruction

In standard BoW, the target is the empirical normalized word-count distribution $y_{\mathrm{BoW}}(x)$, limited to words observed in the document. This produces sparse lexical supervision. In contrast, $y_{\mathrm{DSL}}(x, \pi)$ is dense over $V$ and can assign probability mass to theme-relevant words absent from the document:

$$y_{\mathrm{DSL}}(v \mid x, \pi) > 0 \quad \text{for many } v \notin \mathrm{supp}(x). \quad (15)$$

The reconstruction loss therefore encourages the topic model to explain a semantic distribution rather than only the observed lexical surface form. This distinction explains why DSL can recover thematically relevant words that do not appear in the document, as illustrated in Figure 1.

### J.5. Regularized Implicit-to-Explicit Bayesian Interpretation

Combining the dense semantic reconstruction term with the model-specific regularizer gives the per-document DSL objective

$$\min_\psi \lambda D_{\mathrm{KL}}(y_{\mathrm{DSL}}(x, \pi) \,\|\, \hat{y}_\psi(\cdot \mid x_{\mathrm{emb}})) + \mathcal{R}_\mathcal{M}(x; \psi). \quad (16)$$

For ProdLDA, $\mathcal{R}_\mathcal{M}$ is the variational prior regularizer $D_{\mathrm{KL}}(q_\phi(z \mid x_{\mathrm{emb}})\|p(z))$; for ECRTM and FASTopic, it additionally or alternatively includes the corresponding embedding-clustering or optimal-transport regularization terms. Thus, the first term aligns the explicit topic-model prediction with the LM's implicit semantic posterior predictive target, while the regularizer preserves the structural assumptions of the underlying topic model.

This gives DSL an implicit-to-explicit Bayesian interpretation. The LM supplies an implicit, prompt-conditioned posterior predictive signal over theme-relevant vocabulary words. The topic model then converts this unstructured signal into explicit document-topic proportions and topic-word distributions by matching it within the structured family $\mathcal{P}_\mathcal{M}(x_{\mathrm{emb}})$. As a result, DSL does not use the LM to directly generate topics or hard topic assignments; it distills the LM's thematic posterior predictive signal into a probabilistic topic model that supports interpretation, uncertainty-aware document representations, and retrieval.

## K. Complete Experiment Results

We display the complete automatic topic evaluation results for each number of topics $K = 25, 50, 75, 100$ in Table 10. Across all $K$, DSL consistently improves assignment accuracy, while the strongest backbone depends on the metric: ECRTM + DSL is strongest for $C_V$, ProdLDA + DSL is strongest for LLM rating, and FASTopic + DSL preserves near-perfect I-RBO.

*Table 10.* Automatic evaluation results on the top-15 words for each number of topics ($K = 25, 50, 75, 100$), with five random seeds per $K$. Results not statistically significantly different from the best method ($p \geq 0.05$) under Welch's $t$-test are highlighted in blue. Rows without a suffix use the ProdLDA backbone with our proposed method; rows suffixed with + ECRTM or + FASTopic use the ECRTM or FASTopic backbone.

*(a) $K = 25$*

| Method | 20NewsGroup | | | | TweetTopic | | | | StackOverflow | | | |
|---|---|---|---|---|---|---|---|---|---|---|---|---|
| | $C_V$ | LLM | I–RBO | Purity | $C_V$ | LLM | I–RBO | Purity | $C_V$ | LLM | I–RBO | Purity |
| **Baselines** | | | | | | | | | | | | |
| LDA | .336 | 2.30 | .953 | .296 | .350 | 1.99 | .984 | .416 | .331 | 2.04 | .998 | .142 |
| ProdLDA | .355 | 2.58 | .991 | .335 | .362 | 2.18 | .996 | .509 | .350 | 2.60 | .997 | .238 |
| ZeroShotTM | .354 | 2.54 | .993 | .358 | .361 | 2.42 | .995 | .552 | .362 | 2.80 | .997 | .281 |
| CombinedTM | .357 | 2.64 | .993 | .353 | .364 | 2.43 | .992 | .564 | .366 | 2.86 | .996 | .279 |
| ETM | .339 | 2.43 | .942 | .335 | .348 | 2.30 | .949 | .555 | .350 | 2.35 | .946 | .160 |
| BERTopic | .361 | 2.43 | .988 | .207 | .355 | 2.19 | .996 | .479 | .376 | 2.71 | .994 | .130 |
| ECRTM | .383 | 2.45 | .999 | .335 | .357 | 1.72 | 1.000 | .409 | .367 | 1.93 | 1.000 | .066 |
| FASTopic | .371 | 2.65 | 1.000 | .386 | .260 | 2.07 | .525 | .529 | .363 | 2.64 | .694 | .138 |
| **ProdLDA + DSL** | | | | | | | | | | | | |
| ERNIE-4.5-0.3B | .395 | 2.88 | .995 | .502 | .399 | 2.93 | .992 | .768 | .402 | 2.94 | .992 | .730 |
| Llama-3.1-8B | .376 | 2.94 | .995 | .525 | .392 | 2.91 | .995 | .754 | .399 | 2.97 | .995 | .659 |
| Llama-3.2-1B | .389 | 2.91 | .995 | .543 | .393 | 2.94 | .995 | .770 | .403 | 2.95 | .995 | .678 |
| Qwen-3.5-0.8B | .407 | 2.90 | .986 | .520 | .411 | 2.88 | .979 | .773 | .413 | 2.94 | .989 | .779 |
| Phi-3-mini | .379 | 2.74 | .998 | .536 | .390 | 2.69 | .998 | .767 | .409 | 2.86 | .992 | .790 |
| **ECRTM + DSL** | | | | | | | | | | | | |
| ERNIE-4.5-0.3B + ECRTM | .416 | 2.79 | .987 | .487 | .409 | 2.82 | .984 | .749 | .413 | 2.78 | .993 | .672 |
| Llama-3.1-8B + ECRTM | .398 | 2.80 | .988 | .540 | .392 | 2.75 | .984 | .744 | .396 | 2.98 | .994 | .624 |
| Llama-3.2-1B + ECRTM | .409 | 2.85 | .988 | .567 | .398 | 2.78 | .985 | .764 | .401 | 2.90 | .994 | .625 |
| Qwen-3.5-0.8B + ECRTM | .422 | 2.82 | .986 | .547 | .408 | 2.83 | .983 | .756 | .405 | 2.90 | .992 | .794 |
| Phi-3-mini + ECRTM | .380 | 2.74 | .987 | .519 | .388 | 2.75 | .984 | .749 | .402 | 2.90 | .989 | .739 |
| **FASTopic + DSL** | | | | | | | | | | | | |
| ERNIE-4.5-0.3B | .361 | 2.42 | 1.000 | .483 | .365 | 2.26 | 1.000 | .683 | .395 | 2.61 | 1.000 | .493 |
| Llama-3.1-8B | .370 | 2.24 | 1.000 | .519 | .367 | 2.17 | 1.000 | .691 | .417 | 2.45 | 1.000 | .521 |
| Llama-3.2-1B | .342 | 2.42 | 1.000 | .515 | .359 | 2.20 | 1.000 | .694 | .418 | 2.53 | 1.000 | .522 |
| Qwen-3.5-0.8B | .369 | 2.42 | 1.000 | .474 | .366 | 2.10 | 1.000 | .688 | .404 | 2.66 | 1.000 | .522 |
| Phi-3-mini | .349 | 2.19 | 1.000 | .460 | .360 | 2.14 | 1.000 | .689 | .429 | 2.60 | 1.000 | .524 |

*Table 10.* Automatic evaluation results on the top-15 words for each number of topics, continued.

*(b) $K = 50$*

| Method | 20NewsGroup | | | | TweetTopic | | | | StackOverflow | | | |
|---|---|---|---|---|---|---|---|---|---|---|---|---|
| | $C_V$ | LLM | I–RBO | Purity | $C_V$ | LLM | I–RBO | Purity | $C_V$ | LLM | I–RBO | Purity |
| **Baselines** | | | | | | | | | | | | |
| LDA | .346 | 2.24 | .977 | .299 | .353 | 1.94 | .993 | .425 | .346 | 2.00 | .999 | .167 |
| ProdLDA | .348 | 2.52 | .993 | .357 | .352 | 2.14 | .994 | .532 | .350 | 2.64 | .993 | .266 |
| ZeroShotTM | .357 | 2.58 | .994 | .401 | .351 | 2.31 | .994 | .575 | .365 | 2.86 | .993 | .308 |
| CombinedTM | .349 | 2.58 | .995 | .389 | .356 | 2.36 | .990 | .590 | .364 | 2.84 | .987 | .309 |
| ETM | .343 | 2.35 | .930 | .367 | .354 | 2.20 | .930 | .554 | .349 | 2.30 | .927 | .155 |
| BERTopic | .359 | 2.51 | .992 | .324 | .370 | 2.20 | .996 | .579 | .375 | 2.69 | .996 | .203 |
| ECRTM | .363 | 2.32 | .998 | .378 | .359 | 1.86 | 1.000 | .406 | .377 | 1.93 | 1.000 | .070 |
| FASTopic | .357 | 2.58 | .999 | .424 | .269 | 2.00 | .625 | .557 | .358 | 2.25 | .691 | .166 |
| **ProdLDA + DSL** | | | | | | | | | | | | |
| ERNIE-4.5-0.3B | .379 | 2.84 | .990 | .513 | .388 | 2.88 | .989 | .779 | .395 | 2.90 | .986 | .732 |
| Llama-3.1-8B | .365 | 2.88 | .993 | .556 | .384 | 2.86 | .992 | .773 | .382 | 2.95 | .991 | .704 |
| Llama-3.2-1B | .380 | 2.90 | .991 | .560 | .387 | 2.91 | .992 | .779 | .394 | 2.94 | .991 | .701 |
| Qwen-3.5-0.8B | .403 | 2.85 | .980 | .541 | .398 | 2.89 | .978 | .788 | .403 | 2.88 | .981 | .789 |
| Phi-3-mini | .371 | 2.72 | .994 | .558 | .385 | 2.65 | .995 | .786 | .402 | 2.87 | .988 | .802 |
| **ECRTM + DSL** | | | | | | | | | | | | |
| ERNIE-4.5-0.3B + ECRTM | .407 | 2.86 | .983 | .508 | .398 | 2.81 | .982 | .757 | .414 | 2.91 | .987 | .750 |
| Llama-3.1-8B + ECRTM | .390 | 2.87 | .986 | .567 | .388 | 2.82 | .984 | .758 | .391 | 2.95 | .991 | .740 |
| Llama-3.2-1B + ECRTM | .408 | 2.86 | .980 | .572 | .398 | 2.85 | .983 | .771 | .398 | 2.94 | .992 | .719 |
| Qwen-3.5-0.8B + ECRTM | .428 | 2.86 | .972 | .551 | .406 | 2.82 | .976 | .779 | .403 | 2.89 | .985 | .789 |
| Phi-3-mini + ECRTM | .378 | 2.80 | .988 | .529 | .382 | 2.74 | .986 | .774 | .408 | 2.94 | .988 | .793 |
| **FASTopic + DSL** | | | | | | | | | | | | |
| ERNIE-4.5-0.3B | .343 | 2.28 | 1.000 | .507 | .359 | 2.10 | 1.000 | .703 | .388 | 2.27 | 1.000 | .506 |
| Llama-3.1-8B | .352 | 2.08 | 1.000 | .547 | .353 | 2.02 | 1.000 | .702 | .400 | 2.33 | 1.000 | .532 |
| Llama-3.2-1B | .336 | 2.20 | 1.000 | .536 | .357 | 2.05 | 1.000 | .699 | .392 | 2.35 | 1.000 | .528 |
| Qwen-3.5-0.8B | .345 | 2.17 | 1.000 | .501 | .362 | 2.00 | 1.000 | .693 | .393 | 2.28 | 1.000 | .537 |
| Phi-3-mini | .333 | 2.01 | 1.000 | .499 | .354 | 1.94 | 1.000 | .705 | .392 | 2.28 | 1.000 | .536 |

*Table 10.* Automatic evaluation results on the top-15 words for each number of topics, continued.

*(c) K = 75*

| Method | 20NewsGroup | | | | TweetTopic | | | | StackOverflow | | | |
|---|---|---|---|---|---|---|---|---|---|---|---|---|
| | $C_V$ | LLM | I–RBO | Purity | $C_V$ | LLM | I–RBO | Purity | $C_V$ | LLM | I–RBO | Purity |
| **Baselines** | | | | | | | | | | | | |
| LDA | .341 | 2.19 | .988 | .303 | .350 | 1.95 | .995 | .453 | .354 | 2.05 | .994 | .194 |
| ProdLDA | .351 | 2.46 | .992 | .360 | .355 | 2.09 | .993 | .540 | .355 | 2.63 | .989 | .277 |
| ZeroShotTM | .351 | 2.53 | .994 | .412 | .355 | 2.25 | .993 | .575 | .363 | 2.87 | .991 | .316 |
| CombinedTM | .348 | 2.53 | .992 | .407 | .361 | 2.31 | .987 | .597 | .362 | 2.86 | .983 | .315 |
| ETM | .346 | 2.31 | .930 | .377 | .351 | 2.20 | .932 | .548 | .349 | 2.23 | .930 | .148 |
| BERTopic | .359 | 2.48 | .993 | .425 | .367 | 2.20 | .996 | .589 | .375 | 2.59 | .996 | .223 |
| ECRTM | .353 | 2.25 | .998 | .377 | .354 | 1.89 | 1.000 | .397 | .375 | 1.95 | 1.000 | .056 |
| FASTopic | .353 | 2.55 | .999 | .425 | .283 | 1.94 | .675 | .570 | .367 | 2.20 | .684 | .180 |
| **ProdLDA + DSL** | | | | | | | | | | | | |
| ERNIE-4.5-0.3B | .377 | 2.86 | .990 | .528 | .391 | 2.88 | .988 | .789 | .395 | 2.91 | .983 | .742 |
| Llama-3.1-8B | .360 | 2.85 | .992 | .574 | .381 | 2.87 | .992 | .781 | .383 | 2.95 | .990 | .717 |
| Llama-3.2-1B | .370 | 2.89 | .990 | .572 | .385 | 2.92 | .991 | .788 | .393 | 2.94 | .990 | .711 |
| Qwen-3.5-0.8B | .395 | 2.84 | .977 | .551 | .398 | 2.95 | .975 | .780 | .401 | 2.87 | .981 | .790 |
| Phi-3-mini | .367 | 2.69 | .993 | .563 | .383 | 2.58 | .993 | .793 | .402 | 2.88 | .987 | .808 |
| **ECRTM + DSL** | | | | | | | | | | | | |
| ERNIE-4.5-0.3B + ECRTM | .400 | 2.80 | .985 | .536 | .395 | 2.86 | .982 | .778 | .410 | 2.86 | .986 | .762 |
| Llama-3.1-8B + ECRTM | .385 | 2.83 | .986 | .560 | .383 | 2.85 | .985 | .768 | .395 | 2.94 | .990 | .754 |
| Llama-3.2-1B + ECRTM | .399 | 2.84 | .981 | .590 | .387 | 2.82 | .984 | .783 | .393 | 2.94 | .991 | .738 |
| Qwen-3.5-0.8B + ECRTM | .425 | 2.82 | .970 | .568 | .408 | 2.81 | .974 | .795 | .405 | 2.83 | .980 | .815 |
| Phi-3-mini + ECRTM | .374 | 2.80 | .990 | .542 | .382 | 2.71 | .988 | .789 | .408 | 2.93 | .988 | .815 |
| **FASTopic + DSL** | | | | | | | | | | | | |
| ERNIE-4.5-0.3B | .337 | 2.17 | 1.000 | .522 | .359 | 1.95 | 1.000 | .706 | .378 | 2.14 | 1.000 | .517 |
| Llama-3.1-8B | .336 | 1.99 | 1.000 | .565 | .351 | 1.93 | 1.000 | .707 | .386 | 2.21 | 1.000 | .541 |
| Llama-3.2-1B | .334 | 2.11 | 1.000 | .552 | .355 | 1.97 | 1.000 | .704 | .377 | 2.20 | 1.000 | .535 |
| Qwen-3.5-0.8B | .340 | 2.02 | 1.000 | .516 | .358 | 1.93 | 1.000 | .700 | .383 | 2.15 | 1.000 | .545 |
| Phi-3-mini | .333 | 1.98 | 1.000 | .510 | .350 | 1.91 | 1.000 | .711 | .384 | 2.20 | 1.000 | .542 |

*Table 10.* Automatic evaluation results on the top-15 words for each number of topics, continued.

*(d) K = 100*

| Method | 20NewsGroup | | | | TweetTopic | | | | StackOverflow | | | |
|---|---|---|---|---|---|---|---|---|---|---|---|---|
| | $C_V$ | LLM | I–RBO | Purity | $C_V$ | LLM | I–RBO | Purity | $C_V$ | LLM | I–RBO | Purity |
| **Baselines** | | | | | | | | | | | | |
| LDA | .342 | 2.15 | .991 | .308 | .348 | 1.92 | .995 | .470 | .386 | 1.93 | .915 | .192 |
| ProdLDA | .349 | 2.39 | .992 | .373 | .352 | 2.05 | .992 | .552 | .353 | 2.60 | .986 | .279 |
| ZeroShotTM | .352 | 2.48 | .993 | .417 | .358 | 2.23 | .993 | .589 | .363 | 2.84 | .989 | .324 |
| CombinedTM | .351 | 2.51 | .993 | .415 | .357 | 2.35 | .983 | .599 | .363 | 2.85 | .979 | .320 |
| ETM | .347 | 2.29 | .933 | .361 | .351 | 2.20 | .931 | .548 | .345 | 2.26 | .933 | .141 |
| BERTopic | .359 | 2.48 | .993 | .454 | .364 | 2.19 | .996 | .601 | .370 | 2.53 | .997 | .252 |
| ECRTM | .343 | 2.11 | 1.000 | .364 | .355 | 1.95 | 1.000 | .382 | .371 | 1.98 | 1.000 | .055 |
| FASTopic | .350 | 2.58 | .998 | .431 | .293 | 1.81 | .679 | .571 | .365 | 2.13 | .678 | .199 |
| **ProdLDA + DSL** | | | | | | | | | | | | |
| ERNIE-4.5-0.3B | .373 | 2.86 | .989 | .536 | .389 | 2.89 | .987 | .789 | .395 | 2.90 | .984 | .743 |
| Llama-3.1-8B | .355 | 2.84 | .992 | .580 | .377 | 2.87 | .991 | .786 | .382 | 2.96 | .989 | .720 |
| Llama-3.2-1B | .367 | 2.87 | .989 | .581 | .382 | 2.92 | .990 | .797 | .388 | 2.96 | .990 | .702 |
| Qwen-3.5-0.8B | .391 | 2.84 | .976 | .555 | .397 | 2.81 | .974 | .784 | .397 | 2.86 | .979 | .795 |
| Phi-3-mini | .362 | 2.66 | .993 | .572 | .384 | 2.59 | .992 | .802 | .405 | 2.83 | .986 | .811 |
| **ECRTM + DSL** | | | | | | | | | | | | |
| ERNIE-4.5-0.3B + ECRTM | .393 | 2.82 | .986 | .552 | .390 | 2.80 | .984 | .786 | .405 | 2.84 | .987 | .785 |
| Llama-3.1-8B + ECRTM | .378 | 2.80 | .986 | .578 | .381 | 2.86 | .986 | .789 | .391 | 2.95 | .991 | .761 |
| Llama-3.2-1B + ECRTM | .398 | 2.81 | .979 | .599 | .388 | 2.80 | .985 | .791 | .395 | 2.89 | .991 | .751 |
| Qwen-3.5-0.8B + ECRTM | .419 | 2.80 | .970 | .578 | .403 | 2.82 | .975 | .793 | .410 | 2.84 | .977 | .823 |
| Phi-3-mini + ECRTM | .368 | 2.79 | .991 | .554 | .380 | 2.67 | .991 | .801 | .397 | 2.90 | .989 | .827 |
| **FASTopic + DSL** | | | | | | | | | | | | |
| ERNIE-4.5-0.3B | .333 | 2.08 | 1.000 | .528 | .355 | 1.85 | 1.000 | .715 | .376 | 2.08 | 1.000 | .515 |
| Llama-3.1-8B | .332 | 1.97 | 1.000 | .590 | .351 | 1.91 | 1.000 | .732 | .379 | 2.20 | 1.000 | .575 |
| Llama-3.2-1B | .334 | 2.08 | 1.000 | .559 | .355 | 1.93 | 1.000 | .714 | .373 | 2.21 | 1.000 | .534 |
| Qwen-3.5-0.8B | .335 | 1.97 | 1.000 | .524 | .355 | 1.86 | 1.000 | .700 | .377 | 2.08 | 1.000 | .543 |
| Phi-3-mini | .330 | 1.98 | 1.000 | .526 | .349 | 1.85 | 1.000 | .715 | .374 | 2.15 | 1.000 | .549 |

