# OpenReview forum: "Improving Topic Modeling by Distilling Soft Labels from Language Models"
_ICML.cc/2026/Conference — ICML 2026 regular_

### Official Review · Reviewer_65NJ · 2026-03-04

**Soundness:** 3
**Presentation:** 4
**Significance:** 3
**Originality:** 3
**Overall Recommendation:** 5
**Confidence:** 5

**Summary:**

The paper describes a new self-supervised training routine to distil knowledge from Small Language Models (SLM) into well-known Neural Topic Models (NTP) architectures (in this case ProdLDA) by using the KL-divergence between the logits obtained from the SLM prompted to describe the input text in one word and the probabilities predicted by the NTP. The authors demonstrate superior performance of their approach on most metrics against several baselines across three common datasets. Ablation studies demonstrate the individual contribution of each component of their methodology, where the use of the last hidden state from the SLM as input to the NTP (instead of a general state-of-the-art embedding model) appear to be important for the soft labels reconstruction and the soft labels themselves appear to be important for achieving topics that are more coherent and for assigning topics that are more aligned to ground truth labels.

**Compliance With Llm Reviewing Policy:**

Affirmed.

**Final Justification:**

My justification for the accept score lies not only in the soundness and originality of the approach proposed, but also in the fact that I think this paper has the potential to inform further research on the use of "soft" labels derived from LLMs token probabilities to perform distillation of such generalised models into specialised models (such as a standard VAE topic model in this case). As such I do believe the work deserves an accept, and the authors' response further strengthen this opinion.

**Key Questions For Authors:**

1) What was the criteria for choosing the SLM backbones in the experiments? Have you thought of using different model families as well and/or bigger size LLMs?

2) The I-RBO metric is the only one in which your approach is consistently outperformed by baselines and your approach with BoW targets and no SLM: is this an effect of BoW sparsity and how this relate to the baselines that achieve the highest results under this metric?

3) What was the criteria to use ProdLDA over other similar NTPs? Do you believe your results could be generalisable under different architectures (e.g. substituting the reconstruction loss in one or more architectures you presented as baselines with your soft-label approach)?

4) VAE-based NTP have been recently received less attention than clustering-based and LLM-based. In your work you correctly close a gap between the use of LLMs and VAE NTPs, but I am wondering if your approach could also be used in conjunction with clustering methods like BERTopic (e.g. the distance from clusters in the embedding space could be added as an extra feature during training?)

**Limitations:**

yes

**Strengths And Weaknesses:**

**Strengths**
- *Soundness*: The paper is extremely well justified. The use of the soft labels from the SLM is a very clever way to overcome the classic BoW sparsity problem, with minimal variations required on top of well-understood and sound NTP systems like ProdLDA. The comparison with baselines and across datasets is extensive and inclusive of statistical testing confirming the superiority of the proposed method. The ablation study and prompt sensitivity analysis are also inclusive of statistical testing and complete and give a better understanding of the various elements at play in the proposed methodology.
- *Presentation*: The presentation is extremely clean and the entire methodology is easy to follow and mostly contained in the main text, with only minor details left to appendix. Excellent use of images to summarise the main pipeline and the paper itself appear well organised, with clear main claims and conclusions.
- *Significance*: the paper can potentially lead to a new standard training paradigm for NTP using LLMs logits as soft labels instead of the usual BoW reconstruction targets.
- *Originality*: quite new use of LLMs in the context of NTP, bridging the gap between LLM-based topic modelling and more traditional neural architectures like ProdLDA and, therefore, capitalising on the advantages of both.
*Weaknesses*
- *Soundness*: There are a number of parameters that might affect the final results and that are not analysed in depth, mostly on the side of the SLM, such as prompt (which is correctly shown to be very relevant but it could benefit from a more in depth analysis of different prompting strategies) and layer used to extract input embeddings. Also, different architectures from ProdLDA could have been used for the NTP so as to see whether the method is generalisable to other NTP models. Finally, the choice of SLMs is quite limited and it would be interesting to also have a larger scale analysis of the impact of LLM size on performance (extending the work to LLMs).
- *Presentation*: No weakness to highlight.
- *Significance*: As highlighted above, the paper has the potential of shifting the NTP training paradigm away from the use of BoW targets, but it would require more extensive experimentation to achieve such an impact. Also, such an impact is indeed mostly limited to NTP architectures such as ProdLDA.
- *Originality*: The work is solid, but mostly derivative since the use of soft labels from LLMs is not necessarily new, nor the architectures used here for NTP.

---

> ### Author Rebuttal · Authors · 2026-03-31
>
> We thank the reviewer for the thorough and insightful review. We are encouraged that the reviewer recognize the significance of our work. We will address each question/concern in detail, accompanied by new experiments that we believe will substantively strengthen the paper.
>
> ---
> ## Prompts
>
> While we performed a two-way ANOVA (Section 5.3) which showed that prompt variants has negligible effect on coherence. We agree with the reviewer that a more in-depth analysis of the different prompting strategy such as CoT/few-shot demonstrations. This will be addressed in the final version.
>
> ---
>
> ## Layer Selection
>
> We conducted a small experiments comparing hidden states from different transformer layers as the document input representation on 20NewsGroup with `ERNIE-0.3B`.
>
> | Layer | CV | LLM | I-RBO | Purity |
> |---|---|---|---|---|
> | `GTE-large` | .381 | 2.85 | .988 | .513 |
> | Layer 6 (middle) | .379 | 2.68 | .982 | .501 |
> | Layer 12 (3/4 depth) | .377 | 2.83 | .991 | .512 |
> | Final layer (ours) | **.381** | **2.86** | **.991** | **.520** |
>
> From the table above, we find that replacing the hidden representation to earlier layer resulted in a small but statistically significant drop in performance, which is consistent with our ablation study (Section 5.1) where we replaced the input with contextualized embeddings from `GTE-large` encoder.
>
> ---
>
> ## Generalize to Other NTMs
>
> The reviewer is correct that our method can be generally adapted to our NTMs. We report the detailed results for adaption to ECRTM and FASTopic in the tables below for `ERNIE-0.3B` on 20NewsGroup (averaged across all $K$.
>
> | Model | CV  | LLM | IRBO | Purity  |
> |:---|:---:|:---:|:---:|:---:|
> | ProdLDA | 0.351 | 2.49 | 0.992 | 0.356 |
> | ECRTM | 0.360 | 2.28 | 0.999 | 0.364 |
> | FASTopic | 0.358 | 2.59 | 0.999 | 0.416 |
> | ProdLDA (Ours) | 0.381 | **2.86** | 0.991 | **0.520** |
> | ECRTM (Ours) | **0.404** | 2.82 | 0.985 | **0.521** |
> | FASTopic (Ours) | 0.344 | 2.24 | **1.00** | 0.510 |
>
> This trend holds for the two other datasets. In general, we find that our approach achieve consistent improvements over the baselines and confirms that the benefit is not specific to ProdLDA. We will include the full table in the final version along with a more detailed explanation.
>
> ---
>
> ## Expanded SLM families
> Following the reviewer's suggestion, we have conducted experiments with two additional SLMs, `Qwen3.5-0.8B` and `Phi-3-mini-3.8B`. The table below shows the results on 20NewsGroup using the ProdLDA backbone (averaged across all $K$).
>
> | Model | CV | LLM | I-RBO | Purity |
> |:---|:---|:---:|:---:|:---:|
> | ProdLDA (baseline) | .351 | 2.49 | .992 | .356 |
> | `ERNIE-4.5-0.3B` |  .381 | 2.86 | .991 | .520 |
> | `Llama-3.2-1B` | .375 | **2.87** | .993 | **.559** |
> | `Llama-3.1-8B` | .364 | **2.87** | .993 | **.559** |
> | `Phi-3-mini-3.8B` | .370 | 2.70 | **.994** | .557 |
> | `Qwen3.5-0.8B` | **.399** | 2.86 | .980 | .542 |
>
> In all our experiments, we find that our approach delivers consistent improvements over all baselines regardless of model family.
>
> ---
>
> ### Q1
> We selected smaller LMs due to processing efficiency and to match the size with the contextualized embeddings used in the baselines (`GTE-1.5-0.4B`). From the lack of significant improvements from `Llama-1B` to `Llama-8B`, we don't believe increasing the model size would justify the performance gains.
>
> ### Q2
> The reviewer is correct, our `dense` soft targets assign probability to semantically related words that may not appear in the document, which allows topics to share thematically related vocabulary that improves coherence while slightly reducing I-RBO. To test this, we run a controlled analysis with varying target sparsity by retaining the top-k entries from the soft label distribution. Using `ERNIE-0.3B` on the TweetTopic dataset with $K=50$ we report the results in the table below. For reference BoW has a sparsity rate of 0.3% (5.2 words per doc).
>
>
> | Method | Top-words | CV | LLM | I-RBO | Purity |
> |:---|:---:|:---:|:---:|:---:|:---:|
> | ProdLDA | ~5.2 (BoW) | .352 | 2.14 | .994 | .532 |
> | `ERNIE-0.3B` | 10 | .388 | 2.66 | .998 | .744 |
> | `ERNIE-0.3B`| 50 | .384 | 2.79 | .998 | .771 |
> | `ERNIE-0.3B` | 100 | .380 | 2.79 | .997 | .771 |
> | `ERNIE-0.3B`| 500 | .383 | 2.92 | .996 | .779 |
> | `ERNIE-0.3B` | 1000 | .383 | 2.90 | .992 | .780 |
> | `ERNIE-0.3B` | Full | .388 | 2.88 | .989 | .779 |
>
>
> ### Q3
> We thank the reviewer for pointing this out, we selected ProdLDA as our backbone due to its simplicity. However, as shown in our experiments, our approach successfully transfers to FASTopic & ECRTM. The complete results will be included into the final draft.
>
> ### Q4
> This is a great question. BERTopic's topic representations are derived from class-based TF-IDF scores computed over concatenated cluster documents. Our SLM-derived soft-label distributions could possibly replace or augment this weighting, providing semantic relevance scores rather than purely lexical frequencies for topic word selection.

---

> > ### Author Rebuttal · Reviewer_65NJ · 2026-04-01
> >
> > The authors did a very good job in their rebuttal, including new experiments that address my main questions and concerns about the generalizability of the approach. I therefore increased my score to a full accept.

---

> > > ### Author Response · Authors · 2026-04-01
> > >
> > > We thank you for your thoughtful follow-up and for taking the time to review our rebuttal so carefully. We sincerely appreciate your positive assessment and are especially encouraged that you found our additional experiments and clarifications helpful.
> > >
> > > We are grateful for your constructive feedback throughout the review process. Your questions helped us strengthen both the empirical scope and the presentation of the paper, and we believe the final version will be substantially improved as a result of your suggestions.

---

### Official Review · Reviewer_Py23 · 2026-03-12

**Soundness:** 3
**Presentation:** 4
**Significance:** 2
**Originality:** 2
**Overall Recommendation:** 3
**Confidence:** 4

**Summary:**

This paper proposes a framework that integrates Small Language Models (SLMs) into Neural Topic Models (NTMs) to enhance semantic interpretation. Specifically, the method utilizes the last hidden state representations from SLMs as the input for document encoding and optimizes the NTM by reconstructing the soft logit distributions generated by the SLM. This approach effectively leverages the generative probability distribution of SLMs to guide topic inference. Experimental results demonstrate that the proposed method significantly outperforms baselines in topic coherence, LLM-based evaluations, and clustering purity, while maintaining competitive topic diversity.

**Compliance With Llm Reviewing Policy:**

Affirmed.

**Final Justification:**

I would like to thank the authors for their detailed rebuttal and the additional experiments provided. I acknowledge your explanation regarding the I-RBO metric; the new experiment investigating the impact of vocabulary size provides a much more convincing and coherent justification for the observed variations in topic diversity. In light of these clarifications, I am willing to raise my evaluation score for 'Presentation' to 4.

Despite this clarification, my fundamental concerns regarding the core contributions of this manuscript remain unresolved. The mathematical formulations detailed in the appendix heavily rely on established frameworks, involving merely straightforward variable substitutions rather than introducing novel probabilistic or algorithmic derivations. Consequently, the theoretical depth is distinctly lacking, making the methodological contribution somewhat marginal. Fundamentally, this research operates as a classic "A+B" combination—integrating SLM representations into traditional neural topic models.

While I concede that the empirical performance is solid and the generalizability of the framework is acceptable, ICML stringently prioritizes foundational advancements in machine learning theory and mechanisms. Given its emphasis on empirical combination rather than principled theoretical design, I suggest that this work would be much better suited for an applied machine learning or engineering-oriented academic venue.

**Key Questions For Authors:**

- The experimental results indicate that your model’s performance on the I-RBO metric, a key indicator of topic diversity, is inferior to existing baselines such as ProdLDA, despite the similar underlying architecture. Could you provide a rigorous analysis of why this degradation occurs?
- Figure 1 is currently provided in a low-resolution, non-vector format, lacks sufficient information density despite its large dimensions, and features an erroneous hyperlink to the bibliography. As this figure serves as the primary overview of your proposed framework, please replace it with a professional, high-resolution vector graphic and correct all navigational errors. Enhancing the clarity and quality of this visual representation is essential for improving the overall professional standards of your submission.

**Limitations:**

No, the discussion of limitations is insufficient given the significant degradation in topic diversity observed in the reported I-RBO metrics. To strengthen the paper, the authors should explicitly characterize the source of this diversity loss and provide a comprehensive quantitative analysis of this phenomenon. Furthermore, proposing potential mitigation strategies or discussing the inherent trade-offs between topic coherence and diversity would reflect a more rigorous treatment of the model’s performance boundaries and greatly improve the transparency of the work.

**Strengths And Weaknesses:**

Strengths:
- The proposed framework offers an elegant and effective approach to integrating small language models into neural topic modeling, demonstrating strong practical utility and generalizability.
- The empirical evaluation is extensive and well-structured, providing a robust analysis of performance improvements across multiple datasets, which serves as a solid foundation for the claimed contributions.

Weaknesses:
- The contribution is essentially an empirical "recipe" for performance optimization rather than a fundamental scientific advancement; the paper lacks a deep theoretical investigation into the mechanisms governing the distribution-matching process, rendering it more of an engineering success than a conceptual breakthrough offering novel algorithmic insights.
- The observed degradation in topic diversity, as indicated by the I-RBO metric, is largely left unaddressed, lacking rigorous analysis or mitigation strategies.
- The literature review is incomplete; the paper overlooks relevant work in contextual and graph-based topic modeling, specifically missing *CEMTM: Contextual Embedding-based Multimodal Topic Modeling*.
- There are noticeable formatting inconsistencies, particularly regarding figure presentation (e.g., Figure 1), which detract from the professional quality expected in a top-tier submission.

---

> ### Author Rebuttal · Authors · 2026-03-31
>
> We thank the reviewer for the thoughtful and constructive review. We appreciate the recognition of our framework and our extensive evaluation. Below, we address each concern in detail and outline concrete revisions that we believe strengthen the paper substantially.
>
> ---
>
> ## W1
> Appendix I provides a formal justification of our loss function. Specifically, we show that minimizing our objective (Equation 3) corresponds to a regularized KL projection of the LM-induced semantic distribution onto the constrained hypothesis space of distributions representable by ProdLDA. The reconstruction term finds the closest approximation to the LM's semantic distribution (Equation 8), while the prior regularization term preserves the probabilistic topic structure and prevents latent-space collapse (Equation 9). We will move a condensed version into the main text given the extra space in the final version.
>
> We will additionally discuss how our framework relates to the information bottleneck principle, where BoW reconstruction targets force the topic model to compress documents through a bottleneck that preserves lexical statistics, whereas our soft label targets redirect this compression toward preserving semantic structure. This reframing provides a principled explanation for why our topics better align with human-annotated labels.
>
> ---
>
> ## W2
>
> We thank the reviewer for raising this important point. We provide a rigorous analysis below.
>
> On 20NewsGroup, our best model achieves I-RBO = .993 compared to .999 for ECRTM. This .006 absolute difference means that the overlap between any two topics increases by `fewer than one word on average` (15 words per topic). Since words could be relevant to multiple themes (e.g., "science" is relevant to atheism, physics, and biology), we argue that it is more important for  topics to be semantically meaningful than whether they share zero overlapping words
>
> Additionally, the `I-RBO score from our approach is essentially identical to the ProdLDA family` (see Table below displaying I-RBO scores on all datasets). The "degradation" is only relative to ECRTM and FASTopic, both of which explicitly optimize for diversity.
>
> | Method | 20News | Tweet | Stack |
> |---|---|---|---|
> | ProdLDA | .992 | .994 | .991 |
> | ZeroshotTM | .994 | .994 | .993 |
> | **Ours (Best)** | .993 | .993 | .993 |
>
> Lastly, our method can also be adapted to FASTopic to maintain near perfect diversity. Table below shows the results on TweetTopic using `ERNIE-0.3B`.
> | Model | CV $\uparrow$ | LLM $\uparrow$ | IRBO $\uparrow$ | Purity $\uparrow$ |
> |:---|:---:|:---:|:---:|:---:|
> | FASTopic | 0.276 | 1.96 | 0.626 | 0.557 |
> | FASTopic (Ours) | 0.361 | 2.10 | 1.00 | 0.697 |
>
> However, we find that while it `significantly improves FASTopic`, the ProdLDA backbone achieves the best overall performance trade-off. We find similar trends for the other two datasets.
>
> ---
>
> ## W3
>
> Although we agree with the reviewer, we would like to point out that `CEMTM is already included in Section 2.1`. We will include an extensive list of graph and contextual-based topic model in the final draft given the extra space. We also welcome any suggestions from the reviewer.
>
>
> ---
>
> ## Q1:
> As mentioned in response to W2, the reduction in diversity is statistically insignificant, and further improvements can be achieved by making the dense target sparser. To test this, we run a controlled analysis with varying target sparsity by retaining the top-k entries from the soft label distribution. Using `ERNIE-0.3B` on the TweetTopic dataset with $K=50$ we report the results in the table below. For reference BoW has a sparsity rate of 0.3% (5.2 words per doc).
>
>
> | Method | Top-words | CV | LLM | I-RBO | Purity |
> |:---|:---:|:---:|:---:|:---:|:---:|
> | ProdLDA | ~5.2 (BoW) | .352 | 2.14 | .994 | .532 |
> | ZeroShotTM | ~5.2 (BoW) | .351 | 2.31 | .994 | .575 |
> | Ours | 10 | .388 | 2.66 | .998 | .744 |
> | Ours | 50 | .384 | 2.79 | .998 | .771 |
> | Ours | 100 | .380 | 2.79 | .997 | .771 |
> | Ours | 500 | .383 | 2.92 | .996 | .779 |
> | Ours | 1000 | .383 | 2.90 | .992 | .780 |
> | Ours | Full | .388 | 2.88 | .989 | .779 |
>
> From the table, we see that all generative variants significantly outperform ProdLDA and ZeroShotTM with better I-RBO scores. We also find similar results on the other 2 datasets.
>
> ---
>
>
> ## Q2 + W4:
> We sincerely thank the reviewer for pointing this out, which is an oversight on our part. This was due to inconsistency of the graphing tools used. In the final version, we will (a) replace Figure 1 with a high-resolution vector graphic (PDF/SVG format) to ensure crisp rendering at any zoom level; (b) increase the information density by adding numeric probability values for the most prominent words; (c) fix the erroneous bibliography hyperlink; and (d) conduct a thorough pass for formatting consistency throughout the manuscript. We appreciate the reviewer's attention to presentation quality.

---

> > ### Author Rebuttal · Reviewer_Py23 · 2026-04-03
> >
> > I would like to thank the authors for their detailed rebuttal and the additional experiments provided. I acknowledge your explanation regarding the I-RBO metric; the new experiment investigating the impact of vocabulary size provides a much more convincing and coherent justification for the observed variations in topic diversity. In light of these clarifications, I am willing to raise my evaluation score for 'Presentation' to 4.
> >
> > Despite this clarification, my fundamental concerns regarding the core contributions of this manuscript remain unresolved. The mathematical formulations detailed in the appendix heavily rely on established frameworks, involving merely straightforward variable substitutions rather than introducing novel probabilistic or algorithmic derivations. Consequently, the theoretical depth is distinctly lacking, making the methodological contribution somewhat marginal. Fundamentally, this research operates as a classic "A+B" combination—integrating SLM representations into traditional neural topic models.
> >
> > While I concede that the empirical performance is solid and the generalizability of the framework is acceptable, ICML stringently prioritizes foundational advancements in machine learning theory and mechanisms. Given its emphasis on empirical combination rather than principled theoretical design, I suggest that this work would be much better suited for an applied machine learning or engineering-oriented academic venue.

---

> > > ### Author Response · Authors · 2026-04-04
> > >
> > > We sincerely thank the reviewer for the thoughtful engagement and the raised presentation score. We would like address the remaining concern regarding the nature of our contribution.
> > >
> > > ---
> > >
> > > ## Theoretical Contribution
> > >
> > > We would like to make an argument regarding the theoretical contribution of our work by highlighting two related works (from top ML venues) that uses Bayesian latent variable theory to bridge LLMs and classical probabilistic models [1, 2].
> > >
> > > We follow the formulation of [1] which provided a theoretical framework that LLMs implicitly perform Bayesian inference over latent document-level concepts during next-token prediction:
> > > $$P_{\text{LM}}(w \mid x_{1:t}) = \int_{\Theta} P(w \mid \theta) \, P(\theta \mid x_{1:t}) \, d\theta$$
> > >
> > > This is extended by [2] which frames LLMs as latent variable models with topic variables for demonstration selection, which is similar to our retrieval evaluation.
> > >
> > > We position our work within the theoretical framework of these two papers, both of which established that LLMs implicitly perform Bayesian inference over latent variables during next-token prediction. Our work makes the stronger contribution by performing *explicit* Bayesian inference to decompose the LLM's implicit posterior into structured, interpretable latent topics.
> > >
> > > **Our Formulation**
> > >
> > > We observe that when the LLM is conditioned on a document $x$ with a thematic instruction prompt $p$, the resulting next-token distribution restricted to vocabulary $V$ encodes the LLM's *implicit posterior over document themes*, marginalized onto word space:
> > >
> > > $$y_{\text{target}}(w \mid x) = \text{softmax}\left(\ell_V(x) / \tau\right) \approx \int_{\Theta} P_{\text{theme}}(w \mid \theta)\, P(\theta \mid x, p)\, d\theta$$
> > >
> > > This distribution is dense, semantically coherent, and integrates thematic knowledge from the LLM's entire pretraining corpus by assigning mass to thematically related words absent from the document (Figure 1, green boxes) while suppressing words that appear in the document but are thematically irrelevant (Figure 1, orange boxes).
> > >
> > > We then introduce *explicit Bayesian inference* over a structured latent topic space. Our neural topic model defines a K-component generative model:
> > >
> > > $$P_\phi(w \mid x) = \text{softmax}(\beta^\top \theta_\phi(x))$$
> > >
> > > where $\theta_\phi(x) \in \Delta^{K-1}$ is the document-topic proportion inferred by the encoder, and $\beta \in \mathbb{R}^{K \times |V|}$ is the topic-word matrix. The encoder performs explicit variational inference:
> > >
> > > $$q_\phi(z \mid x_{\text{emb}}) \approx P(z \mid x)$$
> > >
> > > where $z$ is the structured latent topic variable and $x_{\text{emb}}$ is the LLM's hidden state representation (which is a sufficient statistic for $y_{\text{target}}$ since the logits are a linear projection of it).
> > >
> > >
> > > **The theoretical progression across three papers is therefore:**
> > >
> > > - **Xie et al. (ICLR 2022)** [1]: *Existence* — proved that LLMs implicitly perform Bayesian inference over latent concepts. Validated on synthetic data only. No practical method produced.
> > > - **Wang et al. (NeurIPS 2023)** [2]: *Approximation* — showed the latent variable can be approximated as opaque prompt-tuning embeddings (not directly interpretable as topics). Applicable to classification tasks only.
> > > - **Ours**: *Explicit decomposition* — we perform explicit variational Bayesian inference to decompose the LLM's implicit posterior into K structured, interpretable topic components. This yields interpretable topic-word distributions, document-topic proportions, and retrieval-quality representations, validated on 3 real-world datasets with 8 baselines, 6 metrics, and comprehensive statistical testing.
> > >
> > > [1] [An Explanation of In-context Learning as Implicit Bayesian Inference (2022)](https://openreview.net/forum?id=RdJVFCHjUMI)
> > >
> > > [2] [Large Language Models Are Latent Variable Models: Explaining and Finding Good Demonstrations for In-Context Learning (2023)](https://openreview.net/forum?id=BGvkwZEGt7&noteId=iq77U55B2P)
> > >
> > > ---
> > >
> > > We thank the reviewer for encouraging this discussion and for providing constructive criticisms that significantly strengthens our paer. We hope our framing can persuade the reviewer to reconsider our contributions, as it will encourage follow-up works in adapting Bayesian latent variable theories to LLMs.

---

### Official Review · Reviewer_bP1s · 2026-03-12

**Soundness:** 2
**Presentation:** 2
**Significance:** 1
**Originality:** 1
**Overall Recommendation:** 3
**Confidence:** 4

**Summary:**

This paper proposes a novel approach to construct semantically grounded soft-label targets using Language Models (LMs) by projecting next-token probabilities, conditioned on a specialised prompt, onto a predefined vocabulary to obtain contextually enriched supervision signals. Experimental results show that this approach achieves state-of-the-art results.

**Compliance With Llm Reviewing Policy:**

Affirmed.

**Final Justification:**

Slightly improve the rating to acknowledge the authors' hard work in providing more empirical results and analyses. But my major concern exists, which cannot be addressed in the short rebuttal.

**Key Questions For Authors:**

1. In Table 1, why does the smallest model (i.e., ERNIE-4.5-0.3B) achieve the best results compared to Llama-3.2-1B and Llama-3.1-8B? Why does ECRTM achieve the best I-PRO results across three different datasets?

2. In Table 2, why does Llama-1B constantly outperform Llama-8B on Tweet?

**Limitations:**

The paper provides only a brief discussion of the risks of misuse. However, the discussions about the reasons and solutions are very limited. Also, it would be better if the paper included a thorough discussion of technical limitations, e.g., error analyses with error types and cases.

**Strengths And Weaknesses:**

Pros:
1. This paper provides comprehensive experiments, testing and validating the results of their model.


Cons:
1. The motivation is unclear and the writing is poor. First, I am still confused about why we need topic models for LLMs. The introduction says that "Studies that use LLMs for topic generations often involve complex and costly pipelines, which are impractical for
larger corpora. Additionally, prompting-based topic generation methods fail to provide the probabilistic distribution of topics, ..." For topic generations, what if we treat it as a simple generation task? For the probabilistic distribution of topics, what can it do to help us? This paper fails to explain the reasons mentioned above. It just gives something to make up for. Second, what are the disadvantages of existing topic models? What can LLMs provide or solve? As I recall, the short text and sparsity problem is classic, and many researchers have been studying it for a long time.

2. The innovation is insufficient. I am unconvinced by the proposed model as it is too simple and easy. Taking the next-token logits as the topic words distribution lacks design. I am wondering what exactly it learn.

3. The experimental setting is questionable. I am questioning the setting of V. How do you pick out the words? What if you set |V| to a very large or very small number? They all need experiments to validate your settings.

4. The explanation in the experiments is insufficient. To be more specific, although this paper provides comprehensive results, it does not provide clear explanations of the experiments, e.g., in Tables 1 and 2. The essential results, analyses, and explanations are missing in this paper. Therefore, we are unable to understand the reasons behind these experiments.

---

> ### Author Rebuttal · Authors · 2026-03-30
>
> We sincerely thank the reviewer for the detailed feedback and the time invested in evaluating our work. We appreciate that the reviewer acknowledges our novelty and comprehensive experiments. We take this opportunity to address each concern in details, outline concrete revisions and new experiments we have conducted during the rebuttal period, and respectfully clarify several points that we believe may resolve the reviewer's reservations.
>
> ---
>
> ## W1. Why do we need topic models with LLMs?
>
> We would like to clarify that **our goal is not to build topic models for LLMs**, but rather to leverage **SLMs** to improve topic models.
>
> ### 1. Disadvantages of LLM-Based Topic Generation Frameworks
>
> - **Efficiency**: Our approach only requires **a single forward pass using a SLM**, which is extremely efficient compared to all other LLM-based pipelines such as TopicGPT and PromptTopic.
> - **No Probabilistic Distributions**: As mentioned in 2.2, LLM-based generative frameworks **do not produce probabilistic topic distributions**, therefore, these methods cannot be used as document representations for downstream tasks such as retrieval (Section 4.5), and cannot be directly compared with traditional topic models (without topic-word distribution).
> - **Poor Performance**: We run a run TopicGPT pipeline with GPT-4o and **compare the Purity score** with our approach using ERNIE-0.3B with matching number of topics.
>
> | Table 1 | 20 Newsgroups | TweetTopic | StackOverflow |
> |:---|:---:|:---:|:---:|
> | TopicGPT (gpt-4o) | 0.436 | 0.789 | Failed |
> | Ours (ERNIE-0.3B) | 0.535 | 0.820 | 0.737 (avg) |
>
> On the domain-specific StackOverflow dataset, TopicGPT **completely fails** at the keyword extraction phase (extracting variations of the `technology` keyword). Since the LLM processes each short technical title in isolation, it lacks the corpus-wide co-occurrence statistics that allow traditional topic models (and our approach) to distinguish fine-grained differences between domain-specific documents. We will include a short discuss in our final version as it further strengthens our contributions
>
>
> ### 2. Disadvantages of Existing Neural Topic Models
>
> The reviewer correctly notes that data sparsity is a well-studied problem. However, prior solutions have followed the BoW paradigm, either augmenting the input (CombinedTM), or regularizing the topic space (ECRTM, FASTopic).
> Our approach is fundamentally different, rather than patching the sparse BoW signal, we replace the reconstruction target entirely with dense, semantically-grounded supervision derived from the language model's next-token distribution. The ablation study in Table 3 confirms that the soft label target is the single largest contributor to performance gains ("NLL + BoW"), directly validating our central claim.
>
> ---
>
> ## W2. Too simple
>
> We argue the simplicity is the strength of our contribution as our approach can serve as drop-in improvements for any BoW reconstruction-based topic models. We show that our approach can also be adapted to ECRTM and FASTopic with substantial performance gains (with ERNIE-0.3B).
>
> **Table 2: Tweet Topic**
> | Model | CV $\uparrow$ | LLM $\uparrow$ | IRBO $\uparrow$ | Purity $\uparrow$ |
> |:---|:---:|:---:|:---:|:---:|
> | ProdLDA | 0.355 | 2.11 | 0.994 | 0.533 |
> | ECRTM | 0.356 | 1.85 | 1.00 | 0.399 |
> | FASTopic | 0.276 | 1.96 | 0.626 | 0.557 |
> | ProdLDA (Ours) | 0.392 | 2.90 | 0.989 | 0.781 |
> | ECRTM (Ours) | 0.398 | 2.82 | 0.983 | 0.767 |
> | FASTopic (Ours) | 0.361 | 2.10 | 1.00 | 0.697 |
>
> "What exactly does it learn?"
> Our soft label targets represent the latent theme distribution of the document inferred by the language model. By training the topic model to reconstruct this distribution, it learns to approximate the SLM's thematic understanding of each document and assign probability mass to thematically relevant words (Appendix I). We will provide a formal explanation in our final version.
>
> ---
> ## W3. Vocabulary Size
>
> We would like to direct the reviewer's attention to Appendix H (Figure 5), where we have already conducted experiments with $|V| \in {500, 1000, 2000, 4000}$ with consistent improvements at all vocabulary sizes.
>
> ---
> ## W4. Explanation of Results
> Our experiment settings mostly follow prior works (FASTopic, ECRTM). The core finding is that Purity more than doubles on StackOverflow (short text), we refer the reviewer to Appendix F-H for a more thorough analysis of our approach.
>
> ---
>
> ## Questions
>
> Q1: ERNIE-4.5 is a more advanced SLM released in Nov. 2025 with stronger training recipe and target domain alignment. ECRTM achieves a perfect I-RBO due to its ECR loss (forces embeddings apart using OT).
>
> Q2: We attribute this to the nature of the dataset (~28 words) and informal, where larger models can produce flatter soft targets that hedge across related themes, smaller models produce sharper decisive distribution.
>
> ---
>
> We hope our response can resolve the misunderstandings, and alleviate the reviewer's concern regarding our work.

---

> > ### Author Rebuttal · Reviewer_bP1s · 2026-04-04
> >
> > I thank the authors for their rebuttals, which include further experiments and analyses, which help me better understand the empirical results. However, my major concerns about having statistical models in the LLM era remain. Although I agree, it somehow helps smaller models and is more efficient. But there should be better alternatives to achieve this purpose. To encourage the authors, I'd like to increase my score from 2 to 3, but I think they should consider how to motivate better this work given current trends.

---

> > > ### Author Response · Authors · 2026-04-04
> > >
> > > We thank the reviewer for the engagement, and for considering our additional experiments. We would like to address the remaining concerns, which we believe stems from misunderstandings regarding our work.
> > >
> > > ---
> > >
> > > ## Clarifying Misunderstanding
> > >
> > > We believe there is a fundamental misunderstanding about of our method that we would like to respectfully clarify. The reviewer states our method `"somehow helps smaller models and is more efficient"`. We would like to clarify that **our method  uses LLMs (in this case SLMs) to help topic models.** The SLM is solely used to extract soft label targets via a single forward pass. We do not modify or improve the SLM in any way. The goal is to produce a better *topic model*, one that yields more coherent topics, more accurate document-topic assignments, and better retrieval performance than existing methods (LDA, ProdLDA, BERTopic, FASTopic, etc.).
> > >
> > > This distinction is critical is that topic modeling is not a task that LLMs can replace, since they serve fundamentally different purposes. This is because topic modeling is a *corpus-level statistical inference* task that autoregressive LLMs are not designed to perform (showed in our initial rebuttal).
> > >
> > > ---
> > >
> > > ## On "Better Alternatives"
> > >
> > > We believe this comment also stems from the misunderstanding clarified above. If the reviewer is suggesting better alternatives for *improving LLMs*, we agree (e.g., distillation, fune-tuning, etc.). But this is not our goal. Our goal is to improve *topic models*, and we have compared against 8 baselines spanning all major topic modeling paradigms: LDA-family, clustering-based, OT-regularized, and LLM-based topic generation (TopicGPT in rebuttal). Our method outperforms all of them. If the reviewer has specific *topic modeling* methods in mind that we have not compared against, we would welcome the suggestion and are happy to include additional comparisons.
> > >
> > >
> > > ---
> > >
> > > ## Statistical Topic Models in the LLM Era?
> > >
> > > We respectfully argue that having statistical models in the LLM era is **not a dichotomy**. Our work demonstrates that the two are **complementary**, and that statistical topic models offer capabilities that LLMs alone cannot provide:
> > >
> > > **1. LLMs cannot replace topic models for corpus-level analysis**:  Topic models are designed to uncover *shared thematic structure across an entire corpus*. LLMs process documents individually and have no mechanism for discovering corpus-wide structure. As we showed in our rebuttal, TopicGPT (using GPT-4o) completely fails on the domain-specific StackOverflow dataset because it processes each short title in isolation without corpus-wide co-occurrence statistics. This is not an engineering limitation, it is a fundamental architectural constraint of autoregressive LLMs. Our method bridges this gap, where SLM provides per-document semantic understanding, and the topic model aggregates this into global corpus structure.
> > >
> > > **2. Probabilistic topic distributions are essential for downstream applications.** LLM-generated topic labels are discrete, non-probabilistic assignments. Our method produces full probabilistic distributions $\theta(x) \in \Delta^{K-1}$ for every document, which enables retrieval (Table 2), uncertainty modeling, and a more compact representation ($K=50$ vs $d_{\text{model}} = 768+$)
> > >
> > > **3. Scalability.** Our approach requires a single forward pass through a 0.3B SLM (smallest example) to precompute soft targets, after which the topic model trains in minutes on a single GPU. Processing 20,000 documents through an LLM-based pipeline like TopicGPT requires 20,000+ API calls with iterative prompting, costing orders of magnitude more in time and compute.
> > >
> > > ---
> > >
> > > ## Theoretical Motivation: LLMs as Implicit Topic Models
> > >
> > > Beyond the practical arguments, we would like to offer a motivation for *why* combining statistical topic models with LLM/SLM is not merely convenient but has theoretical foundations.  In particular, our work can be framed as *explicit* Bayesian inference decomposing the implicit posterior into structured, interpretable topic components. When the LM is conditioned on a document $x$ with a thematic prompt $p$, the next-token distribution encodes the LM's implicit thematic posterior.
> > >
> > > $$y_{\text{target}}(w \mid x) = \text{softmax}\left(\ell_V(x) / \tau\right) \approx \int_{\Theta} P_{\text{theme}}(w \mid \theta)\, P(\theta \mid x, p)\, d\theta$$
> > >
> > > Our topic model then performs explicit variational inference to *decompose* this into $K$ interpretable components:
> > >
> > > $$P_\phi(w \mid x) = \text{softmax}(\beta^\top \theta_\phi(x)), \quad q_\phi(z \mid x_{\text{emb}}) \approx P(z \mid x)$$
> > >
> > > We refer the reviewer to [our response to reviewer Py23](https://openreview.net/forum?id=S2PARGuQzI&noteId=12bcVseRl2) for a more detailed explanation.
> > >
> > >
> > > ---
> > >
> > > We hope our clarification resolves the misunderstanding about the direction of our contribution. We respectfully ask the reviewer to reconsider the evaluation in light of this clarification.

---

### Decision · Program_Chairs · 2026-04-30

**Decision:**

Accept (regular)

**Comment:**

This paper proposes replacing the standard bag-of-words reconstruction target in neural topic models with semantically grounded soft-label distributions derived from a small language model. The idea is simple but effective, and the empirical evaluation is strong, with consistent improvements across datasets and clear ablations showing that the soft targets drive the gains. The approach is also easy to integrate and computationally efficient.

My justification for the accept score lies not only in the soundness and originality of the approach, but also in its potential to inform future research on using “soft” labels derived from LLM token probabilities to distill general-purpose models into more specialized ones (such as VAEs for topic modeling) as noted by one of the reviewers.

I recommend to improve positioning in the related literature. The paper would benefit from a more complete discussion of related work on incorporating external supervision and LM signals into topic models, including FANToM (NAACL 2025), Labeled LDA, and ANTM (2023). This would clarify that the novelty lies in using LM-induced soft distributions as reconstruction targets rather than in combining LMs and topic models per se.

Overall, despite limited theoretical depth, the paper presents a cool idea with strong empirical support and potential for future impact, not only in topic modeling.